# DAP5 enables main ORF translation on mRNAs with structured and uORF-containing 5′ leaders

Ramona Weber [1,3] ✉, Leon Kleemann [1,4], Insa Hirschberg[2], Min-Yi Chung[1], Eugene Valkov [1,5] & Cátia Igreja [1,6] ✉

Half of mammalian transcripts contain short upstream open reading frames (uORFs) that potentially regulate translation of the downstream coding sequence (CDS). The molecular mechanisms governing these events remain poorly understood. Here, we find that the non-canonical initiation factor Death-associated protein 5 (DAP5 or eIF4G2) is required for translation initiation on select transcripts. Using ribosome profiling and luciferase-based reporters coupled with mutational analysis we show that DAP5-mediated translation occurs on messenger RNAs (mRNAs) with long, structure-prone 5′ leader sequences and persistent uORF translation. These mRNAs preferentially code for signalling factors such as kinases and phosphatases. We also report that cap/eIF4F- and eIF4A-dependent recruitment of DAP5 to the mRNA facilitates main CDS, but not uORF, translation suggesting a role for DAP5 in translation re-initiation. Our study reveals important mechanistic insights into how a non-canonical translation initiation factor involved in stem cell fate shapes the synthesis of specific signalling factors.

The eukaryotic initiation factor (eIF) 4F complex triggers the vast majority of translation initiation events in eukaryotic cells. It is composed of the cap-binding protein eIF4E, the ATP-dependent RNA helicase eIF4A and the scaffolding factor eIF4G which also binds to the poly(A) binding protein (PABP)[1]. Importantly, eIF4G mediates the recruitment of the 43S preinitiation complex (PIC; 40S ribosomal subunit bound to the eIF2:GTP:Met–tRNAi[Met] ternary complex, eIF3, eIF1, and eIF1A) which scans the 5′-UTR of the mRNA in search for an AUG start codon[2].

Mammalian cells express three related eIF4G proteins. Unlike eIF4G1 (hereafter eIF4G) and eIF4G3, death-associated protein 5 (DAP5, eIF4G2) lacks the eIF4E and PABP-binding sites. DAP5 is only homologous to the middle and C-terminal domains of eIF4G (Fig. 1a)

which bind to eIF4A, eIF3 and the β subunit of eIF2 (eIF2β)[3–5]. Thus, DAP5 is a non-canonical initiation factor that directs the ribosome to the mRNA independently of eIF4F.

Most studies indicate that DAP5 stimulates translation in conditions that hinder cap-dependent initiation using internal ribosome entry sites or cap-independent translation enhancers located in the 5′-UTR of specific mRNAs[3,6–13]. Alternatively, DAP5 was proposed to initiate translation via the assembly of cap-bound complexes with proteins other than eIF4E[14,15]. As DAP5 controls the expression of genes required for stem-cell differentiation and embryonic development[16–21], understanding its mode of action is important for elucidating the mechanisms that drive non-canonical translation.

[1]Department of Biochemistry, Max Planck Institute for Developmental Biology, Max-Planck-Ring 5, D-72076 Tübingen, Germany. [2]Friedrich Miescher Laboratory of the Max Planck Society, Max-Planck-Ring 9, D-72076 Tübingen, Germany. [3]Present address: Institute for Regenerative Medicine (IREM), University of Zurich, Wagistrasse 12, CH-8952 Schlieren, Switzerland. [4]Present address: Department of Chemistry, Biochemistry and Pharmaceutical Sciences, University of Bern, Freiestrasse 3, 3012 Bern, Switzerland. [5]Present address: RNA Biology Laboratory & Center for Structural Biology, Center for Cancer Research, National Cancer Institute, Frederick, MD 21702-1201, USA. [6]Present address: Department for Integrative Evolutionary Biology, Max Planck Institute for Biology, Max-Planck-Ring 9, D-72076 Tübingen, Germany. ✉e-mail: ramona.weber@uzh.ch; catia.igreja@tuebingen.mpg.de

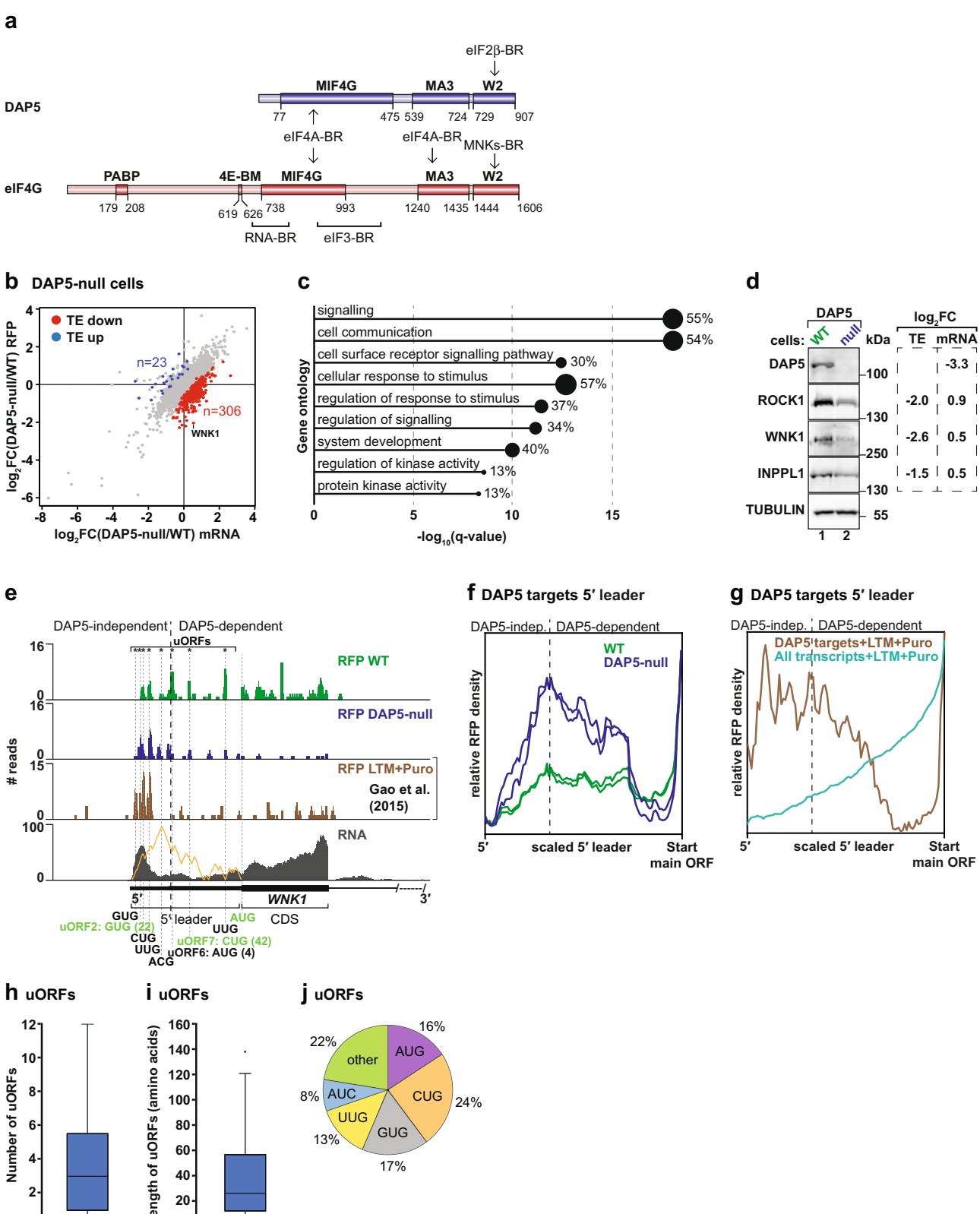

**a**

**b** DAP5-null cells

**c**

**d**

**e**

**f** DAP5 targets 5′ leader

**g** DAP5 targets 5′ leader

**h** uORFs

**i** uORFs

**j** uORFs

Upstream open reading frames (uORFs) are prevalent and translated in the 5′-UTRs (hereafter 5′ leaders) of mammalian mRNAs[22–25]. Expression of downstream and main coding sequences (CDSes) requires scanning of the PIC past the uORFs (leaky scanning) or re-initiation by unrecycled ribosomal complexes after uORF translation[26]. Despite the regulatory roles attributed to uORFs in gene expression and disease[27], the mechanisms controlling uORF

and main CDS translation are incompletely understood. Here, we describe DAP5 as a non-canonical factor that is crucial for the translation of main CDSes in transcripts with distinctively structured 5′ leaders and pervasive uORF translation. Together with eIF4A, DAP5 regulates the translation of mRNAs encoding signalling and regulatory factors with important roles in stem cell and cancer biology, such as kinases and phosphatases. Our findings reveal an

**Fig. 1 | DAP5 mediates the synthesis of signalling proteins. a** Representation of DAP5 and eIF4G. Binding regions (BR) and motifs (BM) for PABP, eIF4E (4E-BM), eIF4A, eIF2β, the mitogen-activated protein kinase (MAPK) interacting protein kinase 1 (MNK1), and MNK2, RNA, and eIF3 are indicated. Domains: middle eIF4G (MIF4G), MA3, and W2. Numbers indicate amino-acid positions. **b** Comparative analysis of translation efficiency (TE) in WT and DAP5-null cells. Genes ($n_{total}$ = 9870) were plotted as a scatter graph according to changes in ribosome occupancy [$log_2$FC RFP] on the $y$ axis and mRNA abundance [$log_2$FC mRNA] on the $x$ axis. Gray: no changes in TE; blue: increased TE; red: decreased TE. **c** Gene ontology terms associated with DAP5 targets. Bar graph shows $-log_{10}$ q values for each over-represented category. Values and black circles indicate the % of genes within each category. **d** Immunoblots were probed with antibodies recognizing DAP5, ROCK1, WNK1, INPPL1, and TUBULIN. Changes in TE [$log_2$FC TE] and mRNA [$log_2$FC mRNA] are depicted next to the blots. **e** Ribosome footprints and total mRNA reads along the *WNK1* 5′ leader and the CDS of exon 1 in WT and DAP5-null cells, and in HEK293 cells treated with lactimidomycin (LTM) and puromycin (Puro)[29]. Predicted propensity for secondary structure is illustrated in orange. uORFs position (*) and length are indicated with the corresponding start codons and highlighted in green when in the same reading frame as the main AUG. DAP5-independent and -dependent translation is indicated with a black dashed line. **f, g** Metagene analyses of ribosome density at the 5′ leaders of DAP5 targets in WT and DAP5-null cells (**f**), or 5′ leaders of DAP5 targets and all transcripts expressed in HEK293 cells treated with LTM and Puro[29] (**g**). DAP5-dependent translation was defined as the position along the 5′ leaders in which RFP density decreases in the absence of DAP5. **h–j** uORF number, length, and start codon usage in DAP5 targets ($n$ = 306 genes). Boxes indicate the 25th to 75th percentiles; black line inside the box represents the median; whiskers indicate the extent of the highest and lowest observations; dots show the outliers. Source data are provided as a Source Data file.

unexpected role for DAP5 in the control of translation in human cells.

## Results

### DAP5 mediates the synthesis of signalling proteins

To study the function of DAP5 in translation, we determined the translational landscape of DAP5-null and wild-type (WT) HEK293T cells using ribosome profiling (Ribo-Seq) and matched transcriptome analysis (RNA-Seq) (Fig. 1b and Supplementary Figs. 1a–f, 2)[22,28]. In the absence of DAP5, a group of genes—DAP5 targets ($n$ = 306 genes)—showed a significant reduction in translation efficiency (TE; ribosome occupancy/mRNA abundance) (Fig. 1b). Although the majority of DAP5-target transcripts were more abundant, the number of ribosomes per mRNA decreased in the null cells (Supplementary Data 1). Other translatome-associated differences included a small cohort of mRNAs with increased TE in the null cells (Fig. 1b, Supplementary Data 1). We also observed pronounced differences in transcript abundance in the null cells (Supplementary Fig. 1d, Supplementary Data 1). These differences may result from effects on transcription and/or mRNA turnover following DAP5 depletion.

DAP5 targets included mRNAs encoding proteins involved in cell signalling, such as the serine/threonine-protein kinases WNK1 [With-No-Lysine (K)1] and ROCK1 (Rho-associated protein kinase 1), the RAC-alpha serine/threonine-protein kinase AKT1 or the phosphatidylinositol 3,4,5-triphosphate 5-phosphatase 2 (INPPL1) (Fig. 1c, Supplementary Data 1). WNK1, ROCK1, and INPPL1 protein levels assessed by immunoblotting were diminished in the absence of DAP5 despite a slight increase in transcript abundance (Fig. 1d, Supplementary Fig. 1g). Decreased protein synthesis in the null cells was not caused by reduced levels of eIF4E, eIF4G, eIF4A and PABP (Supplementary Fig. 1c), or changes in global translation determined using polysome profiles after sucrose density gradient separation (Supplementary Fig. 1h–j). Instead, the association of *WNK1* and *ROCK1* mRNAs with polysomes, but not *GAPDH*, shifted from heavy to light fractions in the absence of DAP5 (Supplementary Fig. 1k–m, lanes 16–18 vs. 12–15). These results indicate that the TE of a specific subset of transcripts is regulated by DAP5.

### DAP5 target mRNA 5′ leaders have unique features

We also observed qualitative changes in the pattern of ribosomal occupancies in DAP5 target mRNAs. Ribosome occupancy at main CDSes was markedly decreased in the absence of DAP5 and skewed towards the 5′ leaders of these transcripts (Fig. 1e, Supplementary Fig. 3). Estimation of footprint density (RFP) in all DAP5 targets revealed that translation was increased on the 5′ leaders in cells lacking DAP5 (Fig. 1f), as measured by the ratio of footprints within the 5′ leader relative to the footprints at the annotated CDS start codon. Translation in the 5′ leaders occurred at uORFs as reflected by experimentally determined quantitative profiling of initiating ribosomes (QTI)[29] (Fig. 1e, g). In QTI, treatment of cells with lactimidomycin

(LTM) and puromycin results in the accumulation of initiating ribosomes at the start codons and in the dissociation of elongating ribosomes, respectively[29].

DAP5 targets have multiple uORFs in the 5′ leader, with a median length of 26 codons, that frequently initiate at near-cognate start codons (CUG, GUG, UUG, and AUC) in addition to the conventional AUG (Fig. 1h–j). *WNK1* showed increased ribosome occupancy in two GUG (one of which is in frame with the main CDS), two CUG, two UUG, one AUG and one ACG uORF (Fig. 1e). These observations suggest that DAP5 mediates CDS but not uORF translation. Close inspection of the RFP profiles revealed that cap-proximal uORF translation is DAP5-independent whereas downstream uORFs and CDS are translated in a DAP5-dependent manner (Fig. 1e, f, Supplementary Fig. 3).

In addition to the presence of uORFs, the 5′ leader sequences of DAP5 targets may form structured elements as they showed increased length, high GC content, and decreased minimum free energy (Supplementary Fig. 4a–c). The increased complexity of the 5′ leader of DAP5 targets was associated with decreased TE of the main CDS (Supplementary Fig. 4d). Moreover, in the null cells the density of RFPs in DAP5-target mRNAs decreased following the predicted structured region in the 5′ leaders of each transcript (Fig. 1e, Supplementary Fig. 3). These observations suggest that structured RNA elements are intrinsically associated with the initiation of translation by DAP5 in uORF-rich transcripts.

### Target mRNA 5′ leaders induce DAP5-dependent translation

We then tested if *WNK1*, *ROCK1,* and *AKT1* 5′ leaders were sufficient to confer DAP5 sensitivity on a *Renilla* luciferase (R-LUC) reporter (Fig. 2a–c). In comparison, R-LUC luminescence driven by *WNK1*, *ROCK1,* and *AKT1* 5′ leaders was reduced in DAP5-null cells to 20%, 30% and 40%, respectively (Fig. 2a–c). Decreased translation of *WNK1*-, *ROCK1*- and *AKT1*-R-LUC reporters was not due to variations in mRNA abundance in the absence of DAP5 (Supplementary Fig. 4e–j). Re-expression of DAP5 (full length; FL) in the null cells restored R-LUC activity (Fig. 2a–d, Supplementary Fig. 4k–m), indicating that the 5′ leaders of *WNK1*, *ROCK1,* and *AKT1* are sufficient to promote DAP5-dependent translation of R-LUC.

### DAP5 MIF4G and W2 domains are required for translation

To elucidate the mechanism of DAP5-dependent translation, we measured the activity of *WNK1*-, *ROCK1-,* and *AKT1*-R-LUC in the null cells upon transient expression of DAP5 mutants. R-LUC activity was not restored if DAP5 was unable to interact with eIF4A (eIF4A*; Fig. 2a–d, Supplementary Fig. 5a). However, binding to eIF4A was not sufficient to induce DAP5-dependent translation as the expression of the eIF4A-interacting domain alone (DAP5 MIF4G) failed to restore R-LUC activity (Fig. 2a–d, Supplementary Figs. 4k–m, 5a). R-LUC activity was also reduced in null cells expressing a DAP5 protein lacking the W2 domain (ΔW2) and unable to associate with the β subunit of the ternary complex (Fig. 2a–d, Supplementary Figs. 4k–m, 5b).

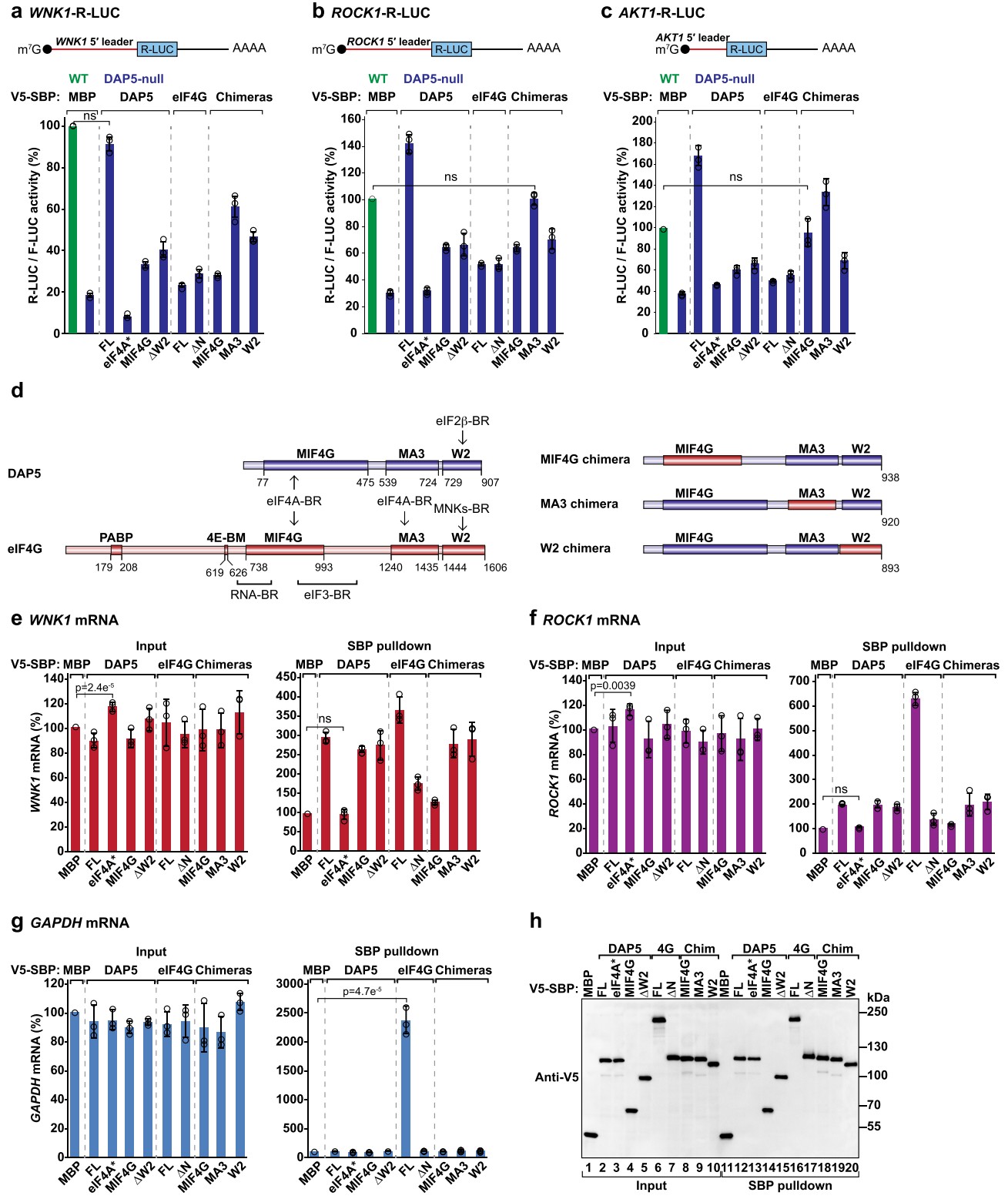

We then asked if overexpression of full-length (FL) or N-terminally truncated eIF4G (lacking the PABP and eIF4E-binding sites; eIF4G ΔN) (Fig. 2d) would suffice to translate the R-LUC reporters in the absence of DAP5. None of the proteins was able to re-establish R-LUC activity (Fig. 2a–d), indicating that *WNK1*, *ROCK1*, and *AKT1* 5′ leaders drive translation of the main CDS in a DAP5-specific manner.

Lastly, we also used DAP5 chimeric proteins where the MIF4G, MA3 or W2 domains were swapped with the respective eIF4G domains

(Fig. 2d). Relative to the re-expression of DAP5 (FL), the MIF4G and W2 chimeras were unable to fully restore R-LUC luminescence in the null cells indicating that these are involved in DAP5-specific interactions and/or functions. The MA3 domain however appears to have similar roles in both eIF4G proteins, since the MA3 chimera still supported R-LUC translation (Fig. 2a–d). All DAP5 protein constructs were expressed at similar levels and mRNA levels were not altered between the conditions (Supplementary Fig. 4e–m).

**Fig. 2 | 5′ leaders determine DAP5-dependent translation of target mRNAs.**
**a−c** WT (green) and DAP5-null (blue) cells were transfected with reporters containing the *WNK1*, *ROCK1*, and *AKT1* mRNAs 5′ leader sequences upstream of R-LUC, and the normalization and transfection control F-LUC-GFP. The plasmids expressing V5-SBP-maltose binding protein (MBP), DAP5 [full length (FL) or the indicated mutants], eIF4G (FL and the indicated mutants), or DAP5-eIF4G chimeric proteins were also present in the transfection mixture. R-LUC activity was quantified, normalized over to that of F-LUC-GFP and set to 100% in WT cells. The mean values ± SD of three independent experiments are shown. Significance was determined by one-way ANOVA test and non-significant (ns) pairs are indicated if $p > 0.05$ (ns). Schematic representations of the reporters are presented above each graph. eIF4A*: eIF4A-binding mutant; MIF4G: DAP5 MIF4G domain; ΔW2: deletion of the DAP5 W2 domain; ΔN: deletion of eIF4G N-terminal region; chimeras: eIF4G MIF4G, MA3, or W2 domains swapped into DAP5. See also Supplementary Fig. 4. **d** Schematic representation of the DAP5, eIF4G, and DAP5-eIF4G chimeras (see

Fig. 1a). **e−g** HEK293T cells were transfected with plasmids expressing V5-SBP-MBP, DAP5 (FL or the indicated mutants), eIF4G (FL or the indicated mutants), or DAP5-eIF4G chimeras. Streptavidin pulldown assays were performed two days post transfection. *WNK1*, *ROCK1*, and *GAPDH* mRNA levels in input (0.8%) and pulldown samples (12%) were determined by quantitative PCR (qPCR) following reverse transcription and set to 100% for V5-SBP-MBP. The mean values ± SD of three independent experiments are shown. The significance of DAP5 or eIF4G binding to mRNA compared to MBP was determined using one-way ANOVA test and indicated significant in the Input panel if $p < 0.05$ or non-significant in the SBP pulldown panel if $p > 0.05$ (ns). **g** Only the binding of eIF4G to the *GAPDH* mRNA in the SBP pulldown panel was found to be significant ($p < 0.05$). **h** Immunoblot depicting the expression and the pulldown efficiency of the V5-SBP-proteins used in **e−g**. Membranes were probed with anti-V5 antibody. Source data are provided as a Source Data file.

## Only eIF4A-bound DAP5 can interact with mRNA

To investigate the recruitment of DAP5 to target mRNAs, we performed RNA-pulldown assays and RT-qPCR. V5-SBP-DAP5 efficiently associated with *WNK1* and *ROCK1* mRNAs but not *GAPDH* (Fig. 2e−h). In contrast, the DAP5-eIF4A* mutant was unable to bind to mRNA (Fig. 2e, f). The MIF4G domain of DAP5 was sufficient to pulldown *WNK1* and *ROCK1* mRNAs, either alone (MIF4G) or when present in other DAP5 constructs (MA3 chimera and W2 chimera; Fig. 2e, f). The DAP5 MIF4G was also specifically required for mRNA binding, as substitution by the respective domain in eIF4G (39% sequence identical) prevented DAP5 recruitment (Fig. 2e, f). Consistent with the role of eIF4A in mRNA binding, we observed that one-third of DAP5 targets showed Rocaglamide A (RocA) sensitivity (Supplementary Fig. 5c; Supplementary Data 2). RocA is a translation inhibitor that clamps eIF4A onto polypurine mRNA sequences[30]. RocA-sensitive mRNAs, such as *WNK1*, show decreased RFP density at the CDS and premature uORF translation in the presence of the drug (Supplementary Fig. 5d, e)[30]. Thus, DAP5 binds to structured mRNAs and stimulates translation when in complex with eIF4A.

The DAP5 ΔW2 protein also bound to *WNK1* and *ROCK1* (Fig. 2e, f), suggesting that the W2 domain does not contribute to target binding. eIF4G bound strongly to all tested mRNAs including the DAP5 targets *WNK1* and *ROCK1*; however, its interaction with mRNA was compromised by the removal of the N-terminal region containing PABP-, eIF4E- and RNA-binding motifs (Fig. 2d−g)[31−34]. All proteins were expressed at equivalent levels and did not alter mRNA input levels (Fig. 2e−g, input panels, h).

Altogether, our findings show that both eIF4G and DAP5 bind to *WNK1* and *ROCK1* mRNAs. DAP5 interaction with the target mRNA is specific and eIF4A-dependent, whereas eIF4G binds to all capped mRNAs as part of the eIF4F complex. We speculate that cap-proximal uORF translation in the structured 5′ leaders of DAP5 targets requires the eIF4F complex while initiation at the main CDS is DAP5- and eIF4A-dependent.

## DAP5-mediated translation is cap-dependent

To understand if the eIF4F complex contributes to DAP5-dependent translation, we overexpressed an improved eIF4E-binding protein (4EBP)[35] in cells and tested binding of DAP5 to *WNK1* and *ROCK1* mRNAs. As shown in cap-based pulldowns, overexpressed 4EBP bound to eIF4E and abolished the interaction with eIF4G (Fig. 3a). Notably, in these conditions that prevent the assembly of the eIF4F complex, binding of V5-SBP-DAP5 to mRNA was suppressed (Fig. 3b, c). Thus, DAP5 participates in cap-dependent translation. All proteins were pulled down at comparable levels in the different experimental conditions (Fig. 3d).

We then studied the requirement for secondary structure in the initiation of translation by DAP5. Analysis of *WNK1* 5′ leader with the G-quadruplex (Gq) secondary structure prediction algorithm QGRS

Mapper[36] identified a motif (292−321 nts) with a high predicted G-score (Fig. 3e). We generated two cap-proximal truncations in *WNK1*-R-LUC mRNA that position the Gq at the 5′ end of the mRNA (Δ1 and Δ2; Fig. 3e). Both truncations reduced mRNA levels, and consequently R-LUC activity (Supplementary Fig. 5f−h), suggesting they might affect mRNA stability and/or transcription. To assess changes only in translation, we determined the protein/mRNA ratios (TE). The *WNK1*-R-LUC mRNA containing the Gq motif directly adjacent to the cap - Δ2 - was not translated (Fig. 3e−g). In agreement with the reports indicating that secondary structures adjacent to the cap prevent 40S subunits from binding to the mRNA[37], addition of an unstructured sequence of 18 CAA repeats to the truncated mRNA restored DAP5-dependent translation of R-LUC (CAA-Δ2; Fig. 3e, g, h, Supplementary Fig. 5j, k). *WNK1*-R-LUC Δ1 mRNA was translated and depletion of DAP5 still reduced R-LUC TE (Fig. 3f, Supplementary Fig. 5f−i). These data support the notion that the 5′ leaders of DAP5 targets contain regulatory and highly structured RNA elements.

We also removed the structured region of the 5′ leader containing the Gq motif (Δstruct) or replaced it with the CAA repeats (Δstruct +CAA; Fig. 3e). In these reporters with 5′ leaders unlikely to adopt strong secondary structures, R-LUC translation did not require DAP5 (Fig. 3i, j, Supplementary Fig. 5l−o). Our results indicate that DAP5 and eIF4A, but not the eIF4F complex, are required to resolve the inhibitory constraints imposed by structured RNA elements present in the 5′ leaders of the targeted transcripts.

## DAP5-dependent translation is regulated by uORF length

Based on the regulatory functions of uORFs in translation[27], we sought to understand their role in the translation of the main CDS by DAP5. The *WNK1* transcript contains at least eight uORFs (Fig. 1e). uORF6 is located downstream of the Gq motif, initiates with a conventional AUG (uAUG) in a different reading frame from the main start codon, is less translated in the absence of DAP5, and encodes a 4 amino acids (aa) peptide (Figs. 1e, 4a). We produced reporters encoding uORF6 with distinct lengths by extending the position of the uSTOP: 118, 30, 19, or 9 codons (Fig. 4a). These *WNK1* reporters were transfected into cells and assayed for R-LUC activity and expression (Fig. 4b−f). Interestingly, DAP5-dependent translation of *R-LUC* was regulated by uORF length. In the presence of a long uORF6 ($uORF_{118}$ and $uORF_{30}$) or $uORF_{19}$, *R-LUC* was poorly translated (Fig. 4c, lanes 4−12 vs 1−3, 4f). In contrast, short uORF6 length ($uORF_9$) primed *R-LUC* translation in a DAP5-dependent manner (Fig. 4c, lanes 13−15, 4f).

Main CDS translation was also inversely correlated with the length of *WNK1* uORF2. This uORF is located upstream of the Gq motif, initiates from a GUG start codon (uGUG) in the same reading frame as the main AUG, and is 22 codons in length (Fig. 1e, Supplementary Fig. 6a). Extension of the position of uORF2 STOP to 29, 39, 49, or 188 codons downstream of an optimized start site (uORF2+: uGUG to uAUG) showed that only short uORF2s ($uORF_{29}$, $uORF_{39}$, and

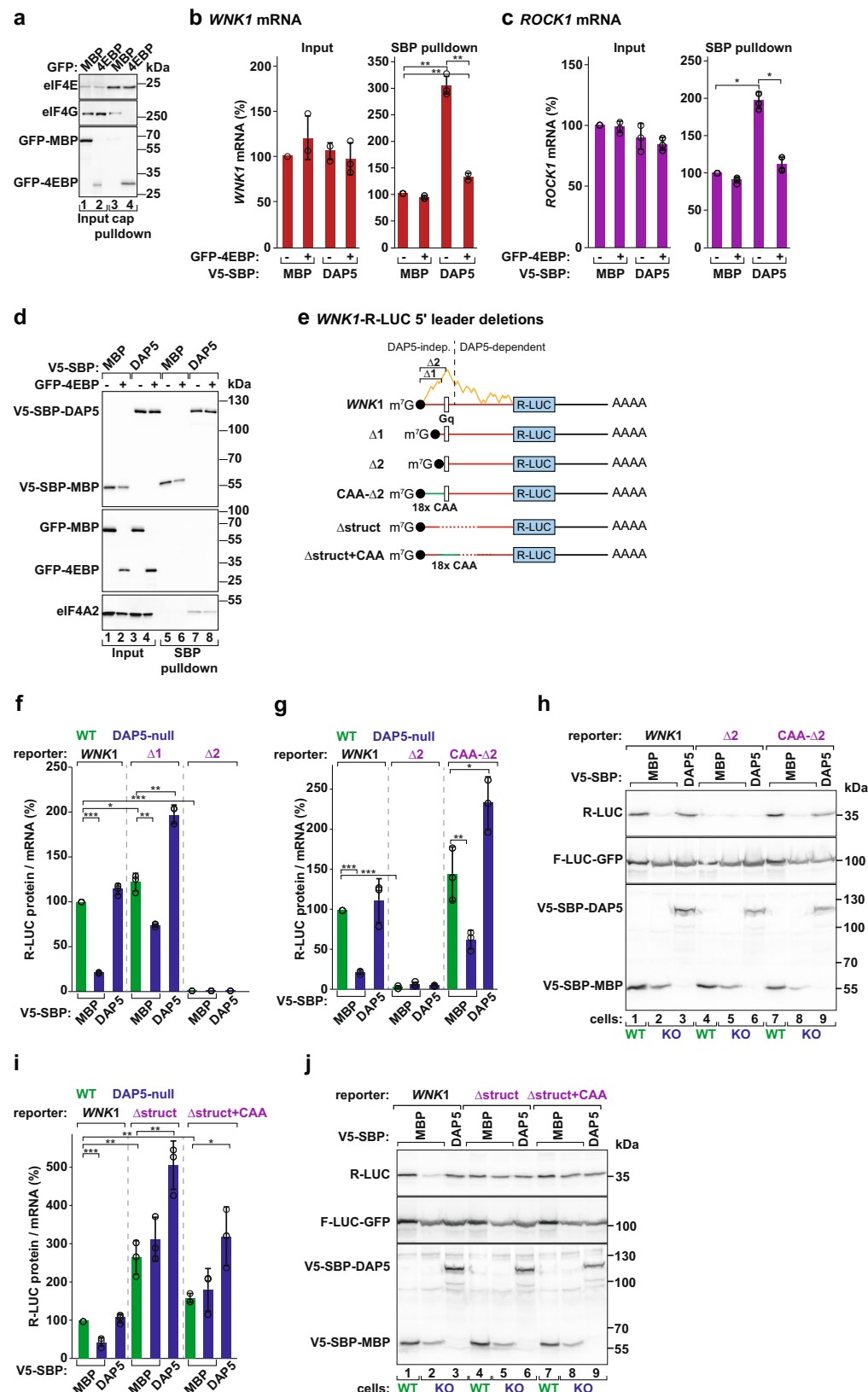

uORF49) sustained efficient DAP5-dependent translation of *R-LUC* (Supplementary Fig. 6a–c). Changes in reporter mRNA abundance were not sufficient to explain the variation in the efficiency of *R-LUC* translation (Supplementary Fig. 6d, e). The finding that long uORFs are more inhibitory to *R-LUC* translation argues for a role of DAP5 in translation re-initiation. This assumption is based on the observation

that ribosomes translating short, but not long uORFs, retain some eIFs (eIF3 subunits, eIF4G and eIF4E) critical for the ability of post-termination ribosomes to avoid recycling, resume scanning, and re-initiate translation at a downstream start codon[38,39].

We also determined the changes in *R-LUC* translation in the absence of uORF translation by altering all cognate and near-cognate

**Fig. 3 | DAP5-dependent translation requires eIF4F-mediated ribosome recruitment. a** Pulldown assay showing the interaction between eIF4E and eIF4G in the presence or absence of GFP-4EBP. Inputs (1% for eIF4E, 0.3% for eIF4G and GFP-tagged proteins) and bound fractions (1% for eIF4E, 2% for eIF4G and GFP-tagged proteins) were analyzed by western blotting. Membranes were probed with anti-eIF4E, eIF4G, and GFP antibodies. **b, c** Binding of V5-SBP-DAP5 or V5-SBP-MBP to *WNK1* and *ROCK1* mRNAs was determined by RNA immunoprecipitation in the presence or absence of GFP−4EBP. mRNA levels in input (0.8%) and IP samples (12%) were quantified by RT-qPCR and set to 100% for V5-SBP-MBP. Bars indicate the mean value; error bars represent SD ($n$ = 3 biologically independent experiments). Significance was determined with one-way ANOVA test and indicated significant if $p < 0.001$ (*) or $p < 1e^{-4}$ (**). **d** Immunoblot depicting the expression of the proteins used in the RNA-IP assay. Inputs were 1% for the V5-SBP-tagged proteins and 0.3% for the GFP-tagged and endogenous proteins. Bound fractions correspond to 1% for the V5-SBP-tagged proteins and 2% for the GFP-tagged and endogenous proteins.

Blots were probed with anti-V5, GFP, and eIF4A2 antibodies. **e** *WNK1*-R-LUC reporters with 5′ leader deletions that partially (Δ1) or completely (Δ2) remove the sequence preceding the quadruple G motif (Gq; open rectangle), and replace it with 18 CAA repeats (CAA-Δ2). Additional reporters lack the Gq motif (Δstruct) or replace it with 18 CAA repeats (Δstruct+CAA). The predicted propensity for secondary structure across *WNK1* 5′ leader is illustrated in orange. **f–j** Cells were transfected with *WNK1*-R-LUC reporters (**e**), F-LUC-GFP, and V5-SBP-MBP or V5-SBP-DAP5. Luciferase activities (protein) were measured and mRNA levels were determined by RT-qPCR. R-LUC values were normalized to F-LUC-GFP. The graphs show the protein-to-mRNA ratios set to 100% in WT cells expressing the *WNK1*-R-LUC reporter. Bars indicate the mean value; error bars represent SD ($n$ = 3 biologically independent experiments). Significance was determined with the one-way ANOVA test and indicated significant if $p < 0.05$ (*), $p < 0.005$ (**), or $p < 5e^{-5}$ (***). Protein levels were also evaluated by immunoblotting (**h** and **j**). See also Supplementary Fig. 5. Source data are provided as a Source Data file.

initiation codons upstream of the main AUG (ΔSTART, Fig. 5a). In agreement with uORF translation limiting the number of ribosomal complexes involved in main CDS translation, we observed a -1.6-fold increase in the TE of *WNK1*-R-LUC reporter (Fig. 5b–d). Importantly, translation of *R-LUC* was still DAP5-dependent (Fig. 5b–e). In the 5′ leaders of the DAP5 targets where scanning is affected by the presence of structured elements, this result suggests that DAP5 does not promote skipping of upstream start codons by scanning PICs that would decrease translation at the main CDS if recognized. That is, the role of DAP5 in translation is independent of leaky scanning.

To rule out the possibility that the intricate nature of the 5′ leader promotes initiation at codons other than the recognizable AUG and near-cognates codons in the *WNK1* ΔSTART-R-LUC reporter, as observed in about 22% of DAP5 targets (Fig. 1j), we also interfered with uORF translation by removing all termination codons in the *WNK1* 5′ leader (ΔSTOP, Fig. 5a). In this reporter mRNA, scanning PICs can initiate at multiple start codons but do not terminate before the main CDS. An N-terminally (N-term) extended version of R-LUC was produced in WT and null cells, i.e., independently of DAP5 (Fig. 5e, lanes 7–9). The molecular weight of this R-LUC indicates that initiation occurred at start sites upstream of the Gq and in frame with the main AUG (other start sites produce undetectable protein products) as a result of the recognition of start codons in the highly structured 5′ leader by slow-moving PICs. The translating ribosomes unwound and moved past the mRNA secondary structures eliminating the requirement for DAP5 and eIF4A. Notably, the short R-LUC (35 kDa) resulting from initiation at the main CDS, and the expected product in a scenario of DAP5-dependent leaky scanning, was not synthesized in WT and null cells (Fig. 5e, lanes 7–9). Thus, we conclude that DAP5-dependent initiation at the main AUG is stimulated by a preceding termination event. We have also generated a *WNK1*-R-LUC mRNA lacking initiation and termination codons in the 5′ leader (ΔSTART/ΔSTOP) (Fig. 5a). None of the R-LUC proteins was observed by Western blotting (Fig. 5e, lanes 10–12) and the TE of the reporter remained low. Our results are consistent with the notion that DAP5 participates in the re-initiation of translation. Such a translation mechanism becomes crucial on long and structured 5′ leaders where the PICs seldom scan until and initiate at the main AUG. We also observed that the TE of the ΔSTART/ΔSTOP reporter (Fig. 5d), and other reporters where R-LUC translation is independent of DAP5 (Fig. 3i), was still improved by the overexpression of DAP5 in the null cells. The fact that DAP5 can still stimulate translation of the main CDS in these transcripts suggests that this non-canonical eIF4G protein can also promote translation in transcripts where re-initiation of translation is not occurring. However, since these transcripts are distinct from the identified DAP5 targets (lack of uORFs or structured RNA elements), it remains unclear if this alternative role of DAP5 can regulate the translation of endogenous mRNAs.

Taken together our findings show that the structured elements in the 5′ leaders of DAP5 targets limit scanning by eIF4F-loaded PICs which tend to recognize upstream start codons and translate uORFs. The inhibitory effects exerted by the structured RNA elements and pervasive uORF translation on main CDS translation are overcome by DAP5 and eIF4A, which may modulate scanning and the re-utilization of ribosomal complexes involved in uORF translation to progressively unwind structured 5′ leaders.

**Simultaneous uORF and main CDS translation in DAP5 targets**

The luciferase-based reporters used in the previous experiments suggest that uORF translation is pervasive and necessary for the DAP5-dependent translation of the main CDS. However, in these experiments, we are unable to detect the synthesis of the short uORF-derived peptides in their natural context, and therefore confirm uORF translation. To simultaneously detect and quantify uORF and main CDS translation, we adopted a split-fluorescent protein approach using mNeonGreen2 (mNG2) that expresses the yellow-green-colored protein in two fragments: mNG2$_{1-10}$ and mNG2$_{11}$. mNG2$_{1-10}$ originates a non-fluorescent mNG2 due to the lack of the 11$^{th}$ β-strand; however, upon co-expression with mNG2$_{11}$ (16 aa peptide), the two fragments assemble a functional mNG2 molecule[40–42]. The uORF2 (22 aa) in the *WNK1* 5′ leader was replaced with the mNG2$_{11}$ CDS initiating with a uAUG. Additionally, the main CDS encoded the EBFP (enhanced blue fluorescent protein) (Fig. 6a). The split-fluorescent reporters were transfected into cells together with a mCherry control reporter.

The non-overlapping excitation and emission spectra of the three fluorophores allowed their simultaneous detection by flow cytometry (Supplementary Fig. 7a–i). Only the co-expression of the two mNG2 plasmids generated the yellow-green fluorescent signal in up to 9% of the cells (Supplementary Fig. 7f, g). Although the complementation efficiency of the split-mNG2 system was low compared to the transfection efficiency in HEK293T cells (-50% in WT cells and -36% in DAP5-null cells, as assessed by the number of mCherry-positive cells, Supplementary Fig. 7i), it clearly showed that uORF translation (mNG2-positive) occurs in the *WNK1* 5′ leader (Fig. 6b, Supplementary Fig. 7g). EBFP fluorescence was only detected in cells expressing the *WNK1*-mNG2$_{11}$- + EBFP reporter (Supplementary Fig. 7h).

Close inspection of the fluorescent output in cells expressing the two mNG2 plasmids showed that the majority of mNG2-positive cells were also EBFP-positive (Fig. 6b), indicating that uORF2 and main CDS are simultaneously expressed. The *WNK1*-mNG2$_{11}$- EBFP reporter also recapitulated DAP5-dependent translation of the main CDS. In the absence of DAP5, a large portion of cells expressing a functional mNG2 (uORF2) do not express EBFP (main CDS) (Fig. 6c, h, j). The number of mNG2 and EBFP double-positive cells was restored upon re-expression of DAP5 in the null cells (Fig. 6d, h, j). Consistent with a block in re-initiation following translation of long uORFs, EBFP fluorescence was

**a** *WNK1*-R-LUC uORF6 reporters

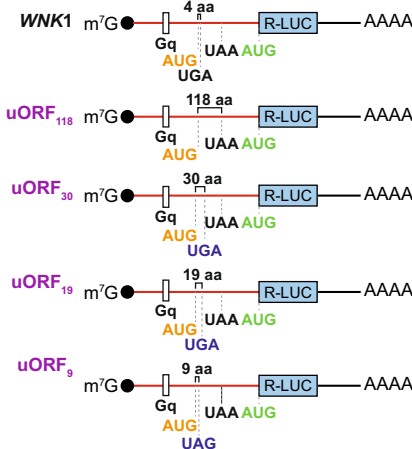

**b** *WNK1*-R-LUC

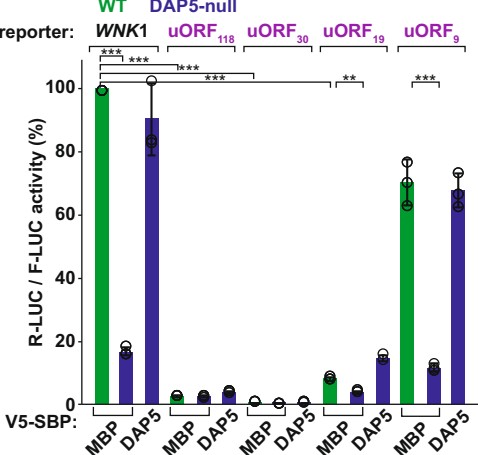

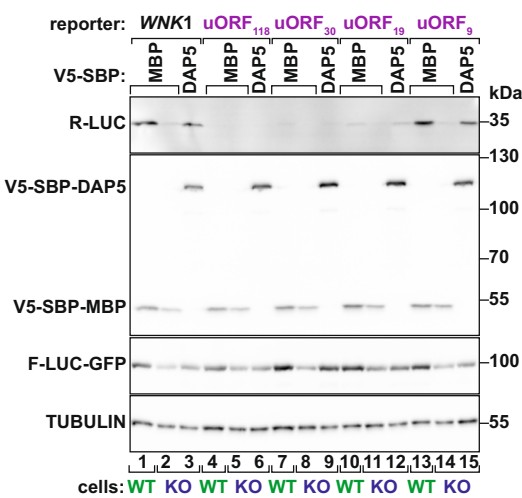

**d** *WNK1-R-LUC* mRNA

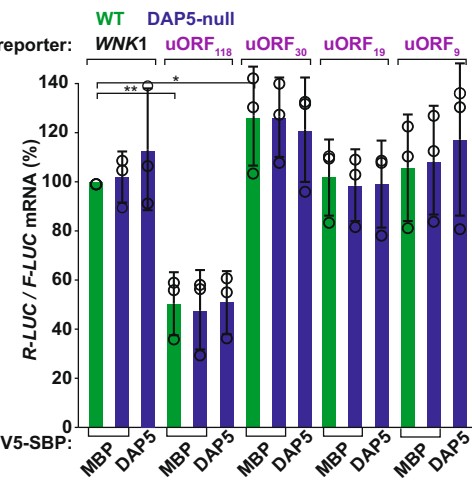

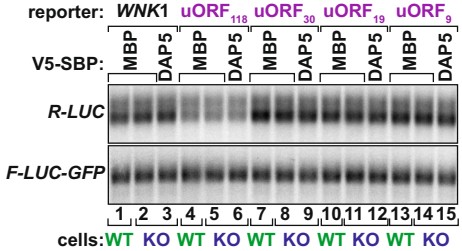

**f** *WNK1*-R-LUC TE

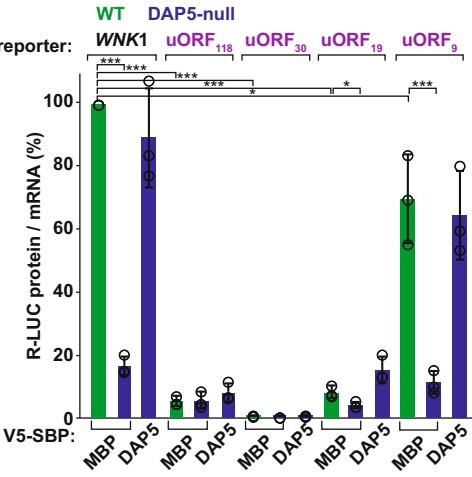

reduced and DAP5-independent in cells expressing the *WNK1* uORF188-mNG2₁₁-EBFP reporter which encodes an mNG2₁₁ peptide fused to 188 amino acids (Fig. 6e–g, i, j). mNG2 expression (uORF translation) did not require DAP5 and was not disturbed by uORF length (Fig. 6k). These observations confirm that DAP5 is not required for uORF translation in the 5′ leader of DAP5 targets but is necessary to promote main CDS translation. Another implication of our results using different reporter systems is that uORF2 sequences and peptides are not relevant for the re-initiation of translation by DAP5, excluding the possibility that uORF-translated peptides influence CDS expression in *cis*. These experiments do not dismiss, however, that *WNK1* uORF-derived peptides are functional in cells.

**Fig. 4 | DAP5-dependent translation is regulated by uORF length. a** Schematic representations of the *WNK1*-R-LUC reporters with changes in uORF6 length. uORF6 initiates from an AUG start codon in the −1 reading frame and encodes a short peptide (four amino acids). uORF$_{118}$: UGA STOP codon was removed changing the length of uORF6 to 118 codons. uORF$_{30}$, uORF$_{19}$, uORF$_{9}$: position of the STOP codon was moved to 30, 19, or 9 codons downstream of aAUG, respectively. The features of the other uORFs in *WNK1* 5′ leader were not modified. **b–f** WT and DAP5-null cells were transfected with different *WNK1*-R-LUC reporters, F-LUC-GFP, and V5-SBP-MBP or V5-SBP-DAP5. Following transfection, luciferase activities (protein) were measured and mRNA levels were determined by northern blotting. R-LUC

values were normalized to the transfection control F-LUC-GFP. The graphs show the luciferase activity (**b**), mRNA levels (**d**) and the protein and mRNA ratios (**f**) in WT and null cells, set to 100% in WT cells expressing *WNK1*-R-LUC. Significance was determined with one-way ANOVA test and indicated significant if $p < 0.05$ (*), $p < 0.005$ (**), and $p < 5e^{-5}$ (***). The immunoblot showing the expression of the different proteins is shown in **c**. TUBULIN served as a loading control. *R-LUC* mRNA levels were determined by northern blotting (**e**), normalized to *F-LUC-GFP,* and set to 100% in WT cells. Bars represent the mean value; error bars represent SD ($n = 3$ biologically independent experiments). Source data are provided as a Source Data file.

### Inhibition of termination impairs DAP5-dependent translation

We also interfered with termination by exploiting a dominant negative mutant of the release factor 1 (eRF1$^{AAQ}$)[43] that causes local translation arrest at STOP codons. eRF1$^{AAQ}$ is unable to hydrolyze the peptidyl-tRNA after STOP codon recognition[44]. Cells were transfected with the *WNK1*-R-LUC (Fig. 7a) and GFP-F-LUC in the absence or presence of increasing amounts of eRF1$^{AAQ}$ and luciferase activities and expression were measured. As expected upon termination inhibition, eRF1$^{AAQ}$ expression decreased R-LUC and GFP-F-LUC protein levels in a concentration-dependent manner (Fig. 7b, lanes 1–4). However, the R-LUC: F-LUC activity ratio varied if R-LUC translation was primed or not by DAP5. In the context of the *WNK1* 5′ leader (*WNK1*-R-LUC and *WNK1*-R-LUC-uORF2 + ), increasing levels of eRF1$^{AAQ}$ proportionally decreased R-LUC activity (Fig. 7a–c). In contrast, DAP5-independent translation of R-LUC using a reporter containing a short 5′ leader (R-LUC) or a *WNK1* 5′ leader without STOP codons (*WNK1*-NO STOP-R-LUC, Fig. 7a), was less affected by the eRF1$^{AAQ}$ mutant. In these cases, R-LUC: F-LUC ratios were constant or even increased in the presence of the mutant release factor (Fig. 7b, c). In all the conditions, R-LUC mRNA levels remained unchanged (Supplementary Fig. 7j, k). These observations suggest that inhibition of termination after uORF translation impairs DAP5-dependent translation of the main CDS.

Similar findings were obtained when 60S recycling was impaired in cells expressing the *WNK1*-R-LUC reporters. shRNA-mediated depletion of the ATP binding cassette sub-family E member 1 (ABCE1 KD; Fig. 7d) decreased the levels of free 60S subunits in cells, as judged in polysome profiles of control (scramble) or ABCE1 shRNA-treated cells (Fig. 7e). In cells with low levels of ABCE1, DAP5-dependent re-initiation of R-LUC translation (*WNK1*-R-LUC and *WNK1*-uORF2 + -R-LUC reporters) was pronouncedly decreased compared to DAP5-independent translation of R-LUC (R-LUC and *WNK1*-NO STOP-R-LUC reporters) (Fig. 7f, g). Depletion of ABCE1 did not affect reporter mRNA levels (Supplementary Fig. 7l, m). Thus, DAP5-dependent main CDS translation is stimulated by an upstream termination event.

We also considered the possibility that the elongating (Fig. 4) or trapped (Fig. 7) 80S ribosomes, could constitute roadblocks to the scanning 43S complexes, and interfere with the function of DAP5 in translation. This interpretation of the data assumes that the 43S complexes scan past the uORF start codons (leaky scanning) and meet the translating or trapped ribosome. However, as shown in Fig. 5, the leaky scanning model does not support DAP5-mediated translation of the main CDS, and the reporters used in this study were optimized to prevent leaky scanning and guarantee initiation at the uORF start site (e.g. uORF2+: the GUG near-cognate start codon was modified to AUG; Supplementary Fig. 6). Thus, we find this interpretation of the data less plausible. Nevertheless, without studies that address uORF and main CDS translation, or 43S and 80S complexes dynamics on a single DAP5-target mRNA, we cannot rule out this last possibility.

### DAP5 targets overlap with re-initiation-dependent mRNAs

Although the molecular mechanisms underlying re-initiation are not well understood, DENR (Density-regulated protein)/MCTS-1 (Malignant T-cell amplified sequence-1) and eIF2D have been shown to

selectively support re-initiation after certain uORFs[45–48]. We compared DAP5 targets with the group of mRNAs showing reduced translation in DENR knockout HeLa cells[45]. Approximately 20% of the DAP5 targets (excluding *WNK1*) were also dependent on DENR for efficient translation (Supplementary Fig. 7n, Supplementary Data 3). mRNAs with significantly reduced TE in the absence of DAP5 or DENR included the proto-oncogenes *c-Raf* and *CDK12*, or the *PI3K* regulatory subunit *R2* (*PI3KR2*). As DENR/MCTS-1 and eIF2D are deacylated tRNA eviction factors[45,49], the significant overlap with DENR targets further implicates DAP5 in non-canonical initiation of translation. This analysis also suggests that DAP5 is not always required for re-initiation as evidenced from the partial overlap with the targets of DENR. For example, translation of *ATF4* a well-characterized re-initiation and leaky scanning dependent mRNA[50], is DAP5-independent (Supplementary Data 1 and 3). Our suspicion is that the absence of recognizable structured elements in the 5′ leader of *ATF4* (and other re-initiation-dependent mRNAs) bypasses the requirement for DAP5 and eIF4A in the re-initiation of translation at the main CDS.

Altogether, our work shows that DAP5 and eIF4A are crucial for main CDS translation on structured mRNAs with pervasive uORF translation and re-initiation events. The data support a model in which structure-triggered binding of the DAP5-eIF4A complex to the mRNA alleviates the energetic barriers modulating the dynamics of scanning and the constrains imposed by uORF translation on main CDS translation.

### Discussion

Here we reveal that DAP5 is a non-canonical factor that mediates translation of the main CDS on mRNAs with structured 5′ leaders and frequent uORF translation. As one of the few initiation factors described to date in the regulation of the translation of re-initiation-dependent transcripts, DAP5 emerges as an important protein in translational control with multiple biological implications. DAP5-dependent transcripts are enriched for regulatory proteins such as kinases and phosphatases, implicating this factor in the control of cell signalling cascades that support cell proliferation and differentiation. Our data, also expands the list of mRNAs in which re-initiation of translation is essential for protein synthesis.

The cues for DAP5-mediated translational control reside in information present in the mRNA 5′ leaders. Transcripts with structure-prone 5′ leaders that lead to pervasive uORF translation selectively require DAP5 for proper translation of the main CDS. These long and burdened sequences restrain scanning of cap-loaded preinitiation complexes, facilitate uORF translation, and limit main CDS translation[51]. Re-utilization of post-termination complexes following uORF translation enables the synthesis of proteins encoded by the main CDS. In this scenario, DAP5 plays a unique role: together with eIF4A it can overcome the increased energetic costs imposed by the structured elements and modulate scanning. We propose that repeated uORF translation and scanning cycles fueled by DAP5 at the scanning-impenetrable 5′ leaders help opening the structure and move the ribosome towards the main CDS (Supplementary Fig. 7o). Approaches that monitor multiple uORF and main CDS translation in

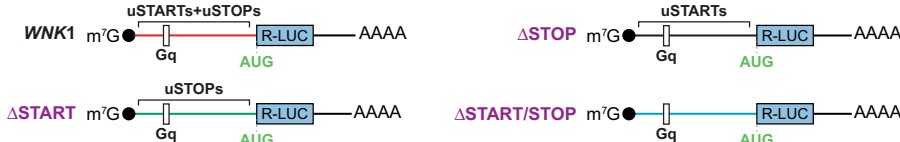

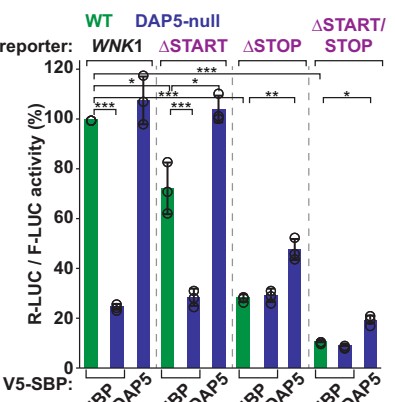

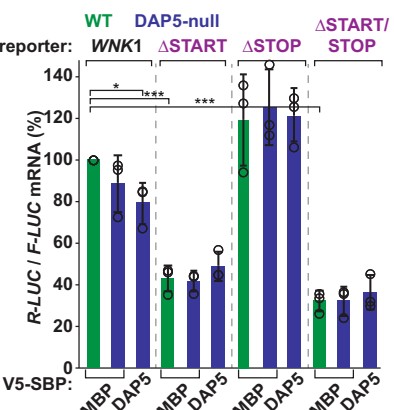

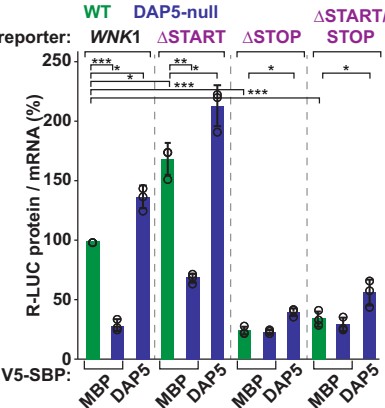

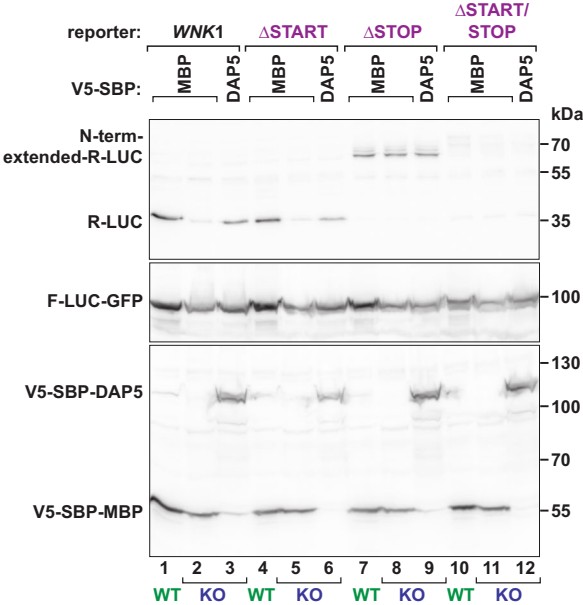

**Fig. 5 | DAP5 does not promote leaky scanning. a** Schematic representations of the *WNK1*-R-LUC reporters without AUG and near-cognate start codons (uSTARTs, ΔSTART), termination codons (uSTOPs, ΔSTOP), or both uSTART and uSTOP codons (ΔSTART/STOP) in the 5′ leader. **b–e** WT and DAP5-null cells were transfected with plasmids expressing *WNK1*-R-LUC reporters, V5-SBP-MBP or V5-SBP-DAP5, and F-LUC-GFP. Following transfection, luciferase activities were measured (**b**) and mRNA levels were determined by RT-qPCR (**c**). R-LUC activity and mRNA levels were normalized to the transfection control F-LUC-GFP and set to 100% in WT cells. Significance was determined by one-way ANOVA test and indicated significant if $p < 0.05$ (*), $p < 0.005$ (**), and $p < 5e^{-5}$ (***). Protein and mRNA ratios in WT and null cells are depicted in **d**. Bars show the mean value and error bars indicate the SD ($n = 3$ biologically independent experiments). The immunoblot in **e** shows the expression levels of the proteins used in the assay. Membranes were incubated with anti-V5, GFP, and R-LUC antibodies. Source data are provided as a Source Data file.

single transcripts will be important to uncover the details of DAP5-dependent translation.

Mechanistically, DAP5 most likely replaces the function of the eIF4F complex. The intricate nature of scanning coupled with the slow translation of sequence biased (GC-rich) uORFs might dissociate or reduce the activity of eIF4F along the long 5′ leaders and

favor binding of DAP5 to 40S subunits. As DAP5 interacts with eIF4A, eIF3, and eIF2β[3–5], its presence on the mRNA may stabilize 40S complexes on the mRNA, license start codon recognition and 80S formation or stimulate a new cycle of scanning and translation. Indeed, DAP5 mutant proteins unable to associate with these initiation factors exhibited reduced ability to promote main CDS

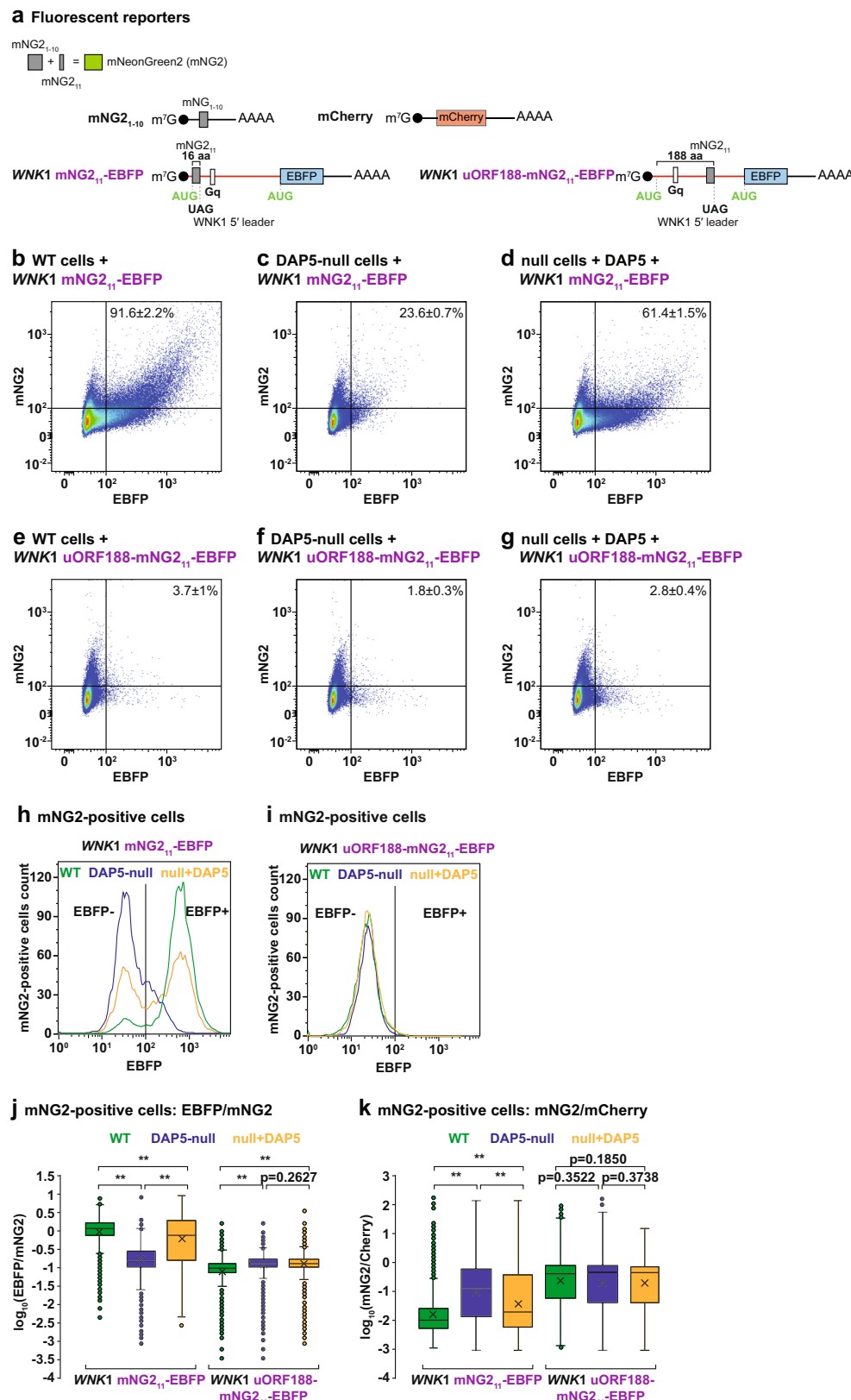

translation, and eIF4G or its N-terminally truncated protein did not substitute DAP5 in null cells. Future studies will enable the detailed characterization of DAP5 functions as a translation initiation factor.

We present additional evidence supporting the role of DAP5 in translational control. Loss of DAP5 only affected the translational efficiency of a fraction of the transcriptome. Ribosome footprint profiles of DAP5 targets and target-based luciferase reporter assays indicated prevalent uORF, but not main CDS translation in the null cells. Increased ribosome density on the uORFs at the expense of the main CDS is consistent with an inability of ribosomes to re-initiate downstream of uORF translation.

**Fig. 6 | Concurrent uORF and main CDS translation in DAP5 targets. a** Schematic representation of the mNeonGreen2 (mNG2) split-fluorescent protein approach and corresponding reporter constructs. Co-expression of the mNG2$_{1\text{-}10}$ and mNG2$_{11}$ fragments originates a functional mNG2 fluorescent molecule[40–42]. mNG2$_{11}$ CDS was inserted in the *WNK1* 5′ leader and replaced uORF2. mNG2$_{11}$ translation initiates at an uAUG in frame with the main CDS and produces a 16 aa protein. Main CDS encoded the EBFP fluorophore. uORF188: The first UAG STOP codon after the uAUG was removed and the mNG2$_{11}$ CDS was inserted next to the UAG STOP located 188 codons downstream of the uAUG. **b–i** WT and DAP5-null cells were transfected with the mNG2$_{1\text{-}10}$, *WNK1*-mNG2$_{11}$-EBFP or *WNK1* uORF188-mNG2$_{11}$-EBFP, mCherry, and V5-SBP-MBP or V5-SBP-DAP5 plasmids. Following transfection, cells were collected and analyzed by flow cytometry. The scattered plots in panels **b–g** show the EBFP and mNG2 signal intensity in all measured cells in the presence (WT cells, null cells +DAP5) or absence (DAP5-null cells) of DAP5. mNG2 and EBFP expression are plotted on a bi-exponential scale and represent ~160,000 cells. The values in the panels represent the proportion of mNG2-positive cells that were also EBFP-positive. **h, i** The histograms show the EBFP signal intensity in mNG2-positive cells (5000 cells in **h** and 3700 cells in **i**) detected in experiments **b–g**. EBFP expression is plotted on a log$_{10}$ scale. **j, k** Box plots of the EBFP/mNG2 and mNG2/mCherry ratios quantified by flow cytometry in WT or DAP5-null cells, and null cells following V5-SBP-DAP5 re-expression. Cells expressed the mNG2$_{1\text{-}10}$, *WNK1*-mNG2$_{11}$-EBFP or *WNK1* uORF188-mNG2$_{11}$-EBFP, and mCherry reporters. Boxes represent the 25th to 75th percentiles; black line shows the median and the cross represents the average; whiskers show the variability outside the upper and lower quartiles; dots show the outliers; $n = 3$ biologically independent experiments. Significance was determined by one-sided Wilcoxon rank-sum test and indicated if $p < 2.2e^{-16}$ (**). null+DAP5: DAP5-null cells re-expressing V5-SBP-DAP5. See also Supplementary Fig. 7.

DAP5 recruitment to the mRNA was determined by the 5′ leader sequence, ribosome loading, and binding to eIF4A. Thus, DAP5 acts upon the initiation of cap-dependent translation on mRNAs that depend strongly on eIF4A for scanning. Understanding the dynamics of eIF4F and DAP5 association/dissociation with the translation machinery will highlight the interplay of different initiation complexes in the synthesis of proteins.

Even though the poor initiation context at the uORFs can lead to frequent leaky scanning in the 5′ leaders of DAP5 targets, our mutational analysis of start and STOP codons showed that translation of short uORFs was mandatory for main CDS expression. DAP5 function was also sensitive to the inhibition of termination and ribosome recycling. In addition, a subset of the DAP5 targets was less translated in cells deficient for DENR, a recycling factor previously implicated in the re-initiation of translation in animal cells[45,47,48,50]. Altogether, our work reports a previously unrecognized role for DAP5 in the control of translation in human cells.

Synthesis of developmental, regulatory, and disease-relevant proteins often occurs on mRNAs with GC- and uORF-rich 5′ leaders that limit the production of proteins that are detrimental to cells if overproduced or deregulated[52–54]. Although the regulatory potential of these 5′ leaders has long been recognized, the molecular mechanisms enforcing translational control are largely unknown. We find that DAP5-dependent re-initiation is required for translation of the main CDS of mRNAs with 5′ leaders where structured regions and uORFs are abundant. DAP5 targets are enriched for mRNAs encoding components of different signalling pathways (kinases, phosphatases, and GTPases) that control cell migration and adhesion, proliferation, differentiation, and transcription. Among the DAP5 targets are members of the WNT pathway with long and structured 5′ leaders, the vascular endothelial growth factor signalling and the MAPK cascade, or different disease-associated genes and proto-oncogenes. Thus, DAP5-dependent translational control of specific signalling components and enzymes that usually have dose-dependent functions efficiently regulates the overall strength of particular signalling pathways in response to stimuli, enabling cells to adapt or adopt different states according to the surrounding environment. Underscoring the physiological importance of DAP5 and re-initiation are the observations that DAP5 deletion in animals results in early embryonic lethality by blocking stem-cell differentiation[16–21]. As several DAP5 targets are known oncogenes and disease-associated genes, future investigations are required to unveil the biological and functional implications of DAP5 in pathological settings. Together with the growing evidence that defective uORF function, polymorphisms, and translational reprogramming at 5′ leaders contribute to various human diseases[27,53,55], our work opens new directions into whether uORF translation, re-initiation, and DAP5 can be exploited for future therapeutic interventions.

Our results also highlight the functional importance of 5′ leaders, uORFs, and re-initiation in the regulation of gene expression. A mechanistic understanding of the influence of alternative 5′ leaders, structured elements, and the increased coding capacity of the genome as a consequence of re-initiation will provide exciting findings on how cells precisely tune protein levels.

## Methods

### Cell lines

All cell lines were cultured at 37 °C and 5% CO$_2$ in Dulbecco's Modified Eagle's Medium supplemented with 10% fetal bovine serum, 2 mM Glutamine, 1× Penicillin and 1× Streptomycin. HEK293T cells were purchased from DSMZ (ACC 635).

### DNA constructs

DNA constructs used in this study are listed in Supplementary Table 1. All the constructs were confirmed by sequencing. To produce the pT7-V5-SBP-C1-MBP, MBP cDNA was introduced in the XhoI and BamHI cut sites of the pT7-V5-SBP-C1 vector. *Hs* DAP5 and eIF4G cDNAs were introduced in the XhoI and KpnI or XhoI and HindIII restriction sites of the pT7-V5-SBP-C1 vector, respectively. To generate the *WNK1*-, *ROCK1*- and *AKT1*-R-LUC reporters, the respective 5′ leader sequences were obtained as synthetic cDNA clones from Invitrogen using the GeneArt tool. Using site-directed mutagenesis, an EcoRI restriction site was inserted upstream of the luciferase ORF present in the pCIneo-R-LUC vector, and EcoRI sites were removed from the *WNK1* and *ROCK1* 5′ leader sequences. The synthetic DNA strings were then inserted into the NheI and EcoRI restriction sites of the modified pCIneo-R-LUC vector. The pSFFV_mNG2(11)1–10 plasmid was a gift from Bo Huang (Addgene plasmid # 82610)[42] and the EBFP-N1 was a gift from Michael Davidson (Addgene plasmid # 54595). The mNG2 1–10 sequence present in pSFFV_mNG2(11)1–10 vector was subcloned into pDNA3.1 using the HindIII and BamHI restriction sites. To generate the EBFP fluorescent reporter with the *WNK1* 5′ leader, the EBFP cDNA was cloned between the EcoRI and XbaI restriction sites of a modified pCIneo-R-LUC, which contains an EcoRI cut site. In this cloning step, EBFP replaces R-LUC ORF. The *WNK1* 5′ leader was then inserted into the NheI and EcoRI sites of the pCIneo-EBFP vector. WNK1-uORF2 was then replaced by the DNA sequence encoding the mNG2 11th β-strand using site-directed mutagenesis. The mCherry sequence was inserted into the XhoI and ApaI restriction sites of the pcDNA3.1-MCS vector. Human eRF1 coding sequence was cloned into the XhoI and HindIII sites of the pλN-HA-C1 vector (Clontech). The pλN-HA-C1-eRF1 plasmid was then used to generate the pλN-HA-C1-eRF1$^{AAQ}$ (G183A G184A) dominant negative mutant by mutagenesis. Human eIF2β cDNA was cloned into the XhoI and BamHI sites of the pT7-EGFP-C1 vector. To generate the DAP5-eIF4G chimeras, DAP5 sequences corresponding to the MIF4G, MA3, and W2 domains were replaced by the respective eIF4G sequences. MIF4G: residues 738–998 of eIF4G iso9 replace residues 78–308 of DAP5. MA3: residues 1240–1435 of eIF4G iso9 replace residues 540–723 of DAP5. W2: residues 1444–1606 of eIF4G iso9 replace residues 730–907 of DAP5. To generate the reporters

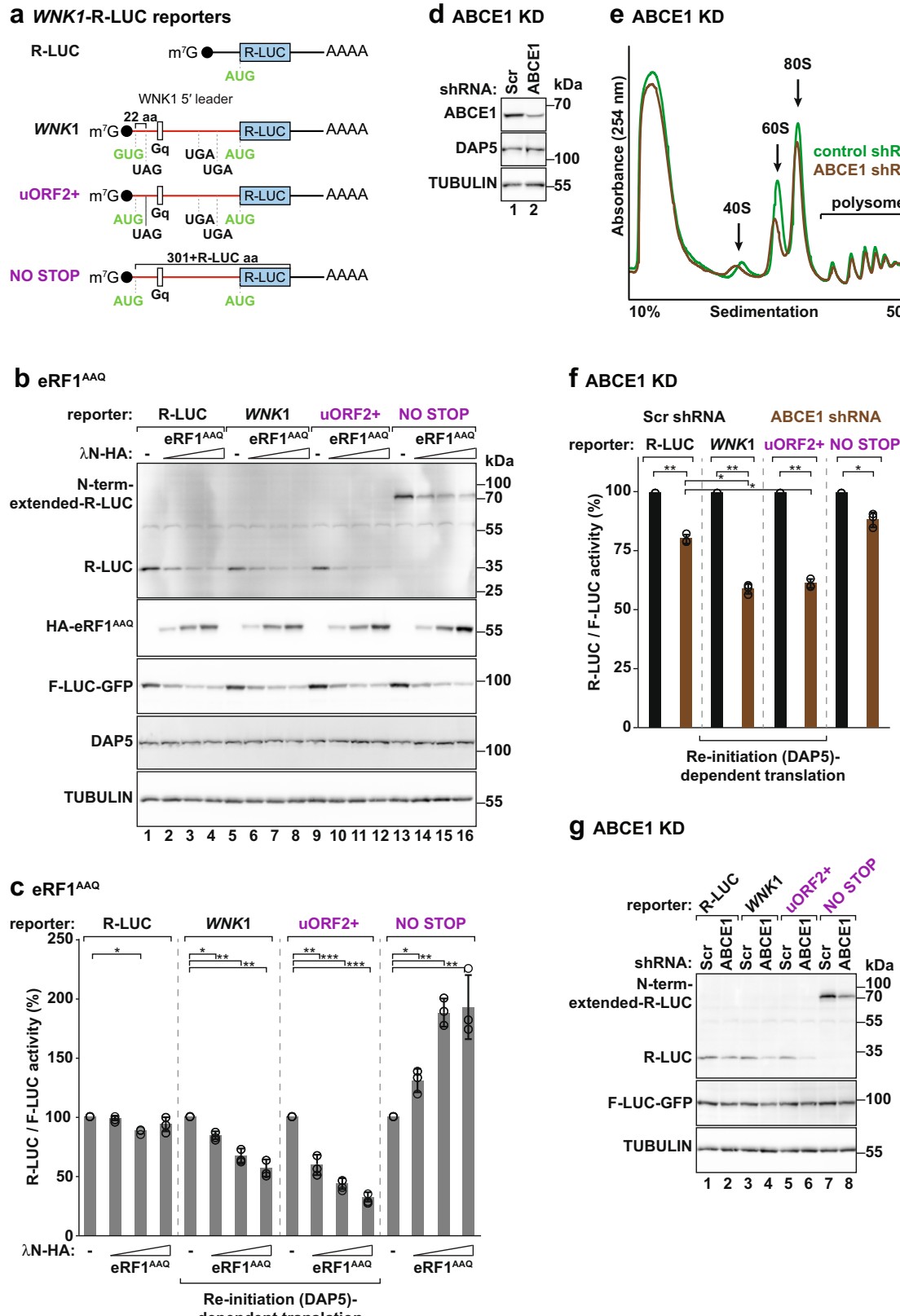

used in Figs. 3–5 and Supplementary Fig. 6, the *WNK1* 5′ leader sequence was modified with the following nucleotide mutations. Δ1: Δ1−217; Δ2: Δ1−266; CAA-Δ2: Δ1−266 replaced by 18× CAA repeats; Δstruct: Δ199−576; Δstruct + CAA: Δ199−576 replaced by 18x CAA repeats; uORF118: A391T; uORF30: A391T, CCC467-469TGA; uORF19: A391T, CGC434-436TGA; uORF9: A391T, TCG404-406TAG; ΔSTART: GTG49-51GAC, GTG93-95GAC, CTG120-122CAC, TTG175-177TAC, ACG275-277AGC, ATG377-379AAC, CTG531-533CAC, TTG845-847TAC; ΔSTOP: G31C, A96T, G161C, A213T, A391T, A444T, G570C, A659T, A733T, A782T, A864T, A868T; ΔSTART/STOP: GTG49-51GAC, GTG93-95GAC, CTG120-122CAC, TTG175-177TAC, ACG275-277AGC, ATG377-379AAC, CTG531-533CAC, TTG845-847TAC, G31C, A96T, G161C,

**Fig. 7 | Inhibition of termination impairs DAP5-dependent translation.**
**a** Schematic representation of *WNK1*-R-LUC reporters with changes in uORF2
length. **b**, **c** HEK293T cells were transfected with the *WNK1*-R-LUC reporters shown
in **a**, F-LUC-GFP and increasing concentrations of λN-HA-eRF1^AAQ. **b** Immunoblot
showing the levels of the expressed proteins. The membranes were blotted with
anti-R-LUC, HA, GFP, DAP5, and TUBULIN antibodies. **c** R-LUC activity was mea-
sured, normalized to F-LUC-GFP and set to 100% in the absence of λN-HA-eRF1^AAQ
for each reporter. Bars indicate the mean value; error bars represent SD ($n = 3$
biologically independent experiments). Significance was determined using the one-
sided ANOVA test and indicated significant if $p < 0.05$ (*), $p < 0.005$ (**), and $p < 5e^{-5}$
(***). See also Supplementary Fig. 7. **d** Western blot showing shRNA-mediated
depletion of ABCE1 in HEK293T cells. TUBULIN served as a loading control. DAP5
expression did not vary in the absence of ABCE1. **e** UV absorbance profile at 254 nM
of scramble shRNA (control, green) and ABCE1-depleted (ABCE1 shRNA, brown)

HEK293T cell extracts after polysome sedimentation in a sucrose gradient. 40S and
60S subunits, 80S monosomes, and polysome peaks are indicated.
**f**, **g** HEK293T cells were treated with scramble (Scr, black) or shRNA targeting *ABCE1*
(brown) mRNA and transfected with the *WNK1*-R-LUC reporters shown in **a**. **f** The
graph shows relative R-LUC activity in control (Scr) and ABCE1 KD cells. R-LUC
activity was normalized to that of F-LUC-GFP and set to 100% in Scr-treated cells for
each reporter. Bars indicate the mean value; error bars represent SD ($n = 3$ biolo-
gically independent experiments). Significance was determined with one-way
ANOVA test and indicated significant if $p < 0.05$ (*) and $p < 0.005$ (**). **g** Immunoblot
illustrating the expression of short and long (N-terminally extended) R-LUC pro-
teins, F-LUC-GFP, and TUBULIN in control and ABCE1-depleted cells. Blots were
probed with anti-R-LUC, GFP, and TUBULIN antibodies. See also Supplementary
Fig. 7. Source data are provided as a Source Data file.

A213T, A391T, A444T, G570C, A659T, A733T, A782T, A864T, A868T;
uORF2+: GTG93-95ATG, TGG111-113ATG; NO STOP: GTG93-95ATG,
TGG111-113ATG, G161C, A659T, A782T; uORF$_{188}$: GTG93-95ATG,
TGG111-113ATG, G161C; uORF$_{49}$: GTG93-95ATG, TGG111-113ATG,
G161C, TCC240-242TGA; uORF$_{39}$: GTG93-95ATG, TGG111-113ATG,
G161C, GTG210-212TAG, uORF$_{29}$: GTG93-95ATG, TGG111-113ATG,
G161C, TCA180-182TGA. The plasmids expressing short hairpin RNAs
(shRNAs) used in the knockdown experiments were derived from the
pSUPERpuro plasmid (a gift from O. Mühlemann) containing the pur-
omycin resistance gene for cell selection. The shRNA target sequences
are listed in Supplementary Table 2.

All the mutants used in this study were generated by site-directed
mutagenesis using the QuickChange Site-Directed Mutagenesis kit
(Stratagene).

### Generation of the DAP5-null cell line
Two sgRNAs targeting DAP5 were designed and cloned into the
pSpCas9(BB)−2A-Puro (PX459) vector [a gift from F. Zhang, Addgene
plasmid 48139[56]] using the CHOPCHOP (http://chopchop.cbu.uib.no)
online tool as previously described[57]. Briefly, HEK293T cells were
transfected with the sgRNA-Cas9 vector. Forty-eight hours later,
edited cells were selected with puromycin (3 µg/ml: Serva Electro-
phoresis). Serial dilutions in 96-well plates were used to isolate
single-cell clones. Genomic DNA was extracted from the different
clones using the Wizard SV Genomic DNA Purification System (Pro-
mega). The DAP5 locus was PCR amplified and Sanger sequencing of
the targeted genomic regions indicated two frameshift mutations in
exon 9 (172 bp deletion in exon/intron 10, and a 1 bp insertion) tar-
geted by sgDAP5-a (Supplementary Fig. 2a). These mutations caused
defective splicing and intron retention, as evidenced by subsequent
RNA sequencing (Supplementary Fig. 1b). Two mutations were
detected in exon 11 (1 bp insertion and 12 bp deletion) targeted by
sgDAP5-b. The lack of DAP5 protein was further confirmed by wes-
tern blotting (Fig. 1d, Supplementary Fig. 1c). RNA sequencing
revealed that DAP5 transcript levels were severely reduced in the null
cells compared to wild-type cells (Supplementary Fig. 1a), most likely
as a result of non-sense mediated decay. The following guide
sequences were used: sgDAP5-a: 5′-CACGTACCTTGGCTCGTTCA-3′;
sgDAP5-b: 5′-ACACCATTGGGTTCCTCGCA-3′.

### Ribosome profiling and RNA sequencing
For ribosome profiling and RNA sequencing HEK293T wild-type and
DAP5-null cells were plated on 10 cm dishes 24 h before harvesting
($3.2 \times 10^6$ WT cells and $3.5 \times 10^6$ null cells per plate). Cells were har-
vested as described in Calviello et al.[58]. Importantly, cells were not
incubated with cycloheximide before harvesting. Cycloheximide
(100 µg/ml, Serva Electrophoresis) was only present in the washing
and lysis buffer, as described in Calviello et al. (2016)[58]. For total RNA
sequencing, RNA was extracted using the RNeasy Mini Kit (50) (Qia-
gen) and processed according to the Illumina TruSeq RNA Sample Prep

Kit. For ribosome profiling the original protocol[59] was used in a mod-
ified version also described in Calviello et al.[58]. The ribosome profiling
and total RNA sequencing pools were sequenced on an Illumina
Hiseq3000 instrument. Reads originating from ribosomal RNA were
removed using Bowtie2[60]. Remaining reads of the RNA sequencing
library were mapped onto the human genome using Tophat2[61] which
resulted in 15.7–20.5 million mapped reads with an overall read map-
ping rate >94% for the RNA sequencing experiment. Ribosome profil-
ing reads were subjected to statistical analysis using RiboTaper that
aims at identifying actively translating ribosomes based on the char-
acteristic three-nucleotide periodicity[58]. Reads of 29 and 30 nucleo-
tides length showed the best three-nucleotide periodicity and where
therefore used for subsequent mapping onto the human genome. This
resulted in 2.8–3.8 million mapped reads with an overall read mapping
rate >95% for the ribosome profiling experiment. Read count analysis
was performed using QuasR[62]. Differential expression analysis was
conducted using edgeR[63,64]. Translation efficiency (TE) was calculated
using RiboDiff[28].

Harringtonine, LTM, and QTI datasets from human HEK293 cells
were downloaded from the Sequence Read Archive database with the
accession number SRA056377 and SRA160745. RocA and DENR data-
sets were retrieved from the GEO database accession numbers
GSE70211 and GSE140084, respectively. Ribosomal RNA reads were
filtered using Bowtie 2[60]. The remaining reads were mapped on the
hg19 (UCSC) human genome or the mm9 (UCSC) mouse genome with
TopHat2[61]. No specific filters for read length were applied.

### Analysis of GO terms and nucleotide compositions
Upregulated and downregulated gene groups were defined as being
significantly deregulated (false discovery rates; FDR < 0.005) with a
$log_2FC > 0$ and $log_2FC < 0$, respectively. No cut-off of the logFC value
was applied so that genes with little but significant changes could also
be detected. GO analysis was performed with the R based package
goseq[65]. For analysis of 5′ leader nucleotide composition, the respec-
tive mRNA sequences were fetched using biomaRt[66,67]. Analysis of GC
content and length of 5′ leader was performed with R-based scripts.

RNA structures were calculated using the ViennaRNA package
2.0[68]. Metagene analysis was performed using the Deeptools suite of
functions[69]. For uORF number, size, and start codon analysis the
accumulation of ribosome footprint on start codons was assessed
using the ribosome profiling dataset in HEK293 cells treated with
harringtonine[23]. Identity of the start codon and the corresponding
STOP codon was manually assigned.

Ribosome footprint density plots for individual sequencing tracks
were visualized using the Integrative Genomics Viewer visualization
tool[70,71].

### Transfections, northern and western blotting
In the rescue assays described in Figs. 2–6, $0.64 \times 10^6$ WT cells or
$0.7 \times 10^6$ null cells were transfected, after seeding in 6-well plates, using

Lipofectamine 2000 (Invitrogen). The transfection mixtures contained different amounts of the plasmids expressing R-LUC, GFP-F-LUC or V5-SBP-fusion proteins (WNK1-R-LUC reporters: 0.5 µg; GFP-F-LUC: 0.25 µg; V5-SBP-MBP: 0.3 µg; V5-SBP-DAP5 FL and MIF4G: 0.8 µg; V5-SBP-DAP5-eIF4A*: 3.25 µg; V5-SBP-DAP5 ΔW2: 1.2 µg; V5-SBP-eIF4G FL: 3.25 µg; V5-SBP-eIF4G ΔN: 0.8 µg; V5-SBP-Chimeras: 0.8 µg). For the experiment shown in Fig. 7, λN-HA-eRF1 G183A G184A was titrated using 0.25 µg, 0.75 µg, and 1.25 µg of plasmid DNA.

Cells were harvested two days after transfection and firefly and *Renilla* luciferase activities were measured using the Dual-Luciferase reporter assay system (Promega). Total RNA was isolated using TriFast (Peqlab biotechnologies). For northern blotting, total RNA was separated in 2% glyoxal agarose gels and blotted onto a positively charged nylon membrane (GeneScreen Plus, Perkin Elmer). [$^{32}$P]-labeled probes specific for each transcript were generated by linear PCR. Hybridizations were carried out in hybridization solution (0.5 M NaP pH = 7.0, 7% SDS, 1 mM EDTA pH = 8.0) at 65 °C overnight. After extensive washes with washing solution (40 mM NaP pH = 7.0, 1% SDS, 1 mM EDTA pH = 8.0), the membranes were exposed and band intensities were quantified with a PhosphoImager.

Western blot was performed using standard methods. In brief, cells were washed with PBS and lysed with sample buffer (100 mM Tris-HCl pH = 6.8, 4% SDS, 20% glycerol, 0.2 M DTT) followed by boiling 5 min at 95 °C and vortexing to shear genomic DNA. After SDS-PAGE, proteins were transferred onto a nitrocellulose membrane (Santa Cruz Biotechnology) by tank transfer. Primary antibodies were incubated overnight at 4 °C and secondary antibodies for an hour at room temperature. All western blots were developed with freshly mixed 10 A: 1B ECL solutions and 0.01% H$_2$O$_2$ [Solution A: 0.025% Luminol (Roth) in 0.1 M Tris-HCl pH = 8.6; Solution B: 0.11% P-Coumaric acid (Sigma Aldrich) in DMSO]. Antibodies used in this study and corresponding dilutions are listed in Supplementary Table 3.

### Reverse transcription (RT) and quantitative PCR (qPCR)

1 µg of RNA was mixed with 0.66 µg of random hexamer primers (N$_6$) and denatured at 72 °C for 5 min. After addition of a reaction mixture containing a final concentration of 1× RT buffer, 20 U RiboLock RNase Inhibitor (Thermo Scientific), and 1 mM dNTPs, the RNA samples were incubated at 37 °C for 5 min. Incubation with RevertAid H Minus Reverse Transcriptase (200 U, Thermo Scientific) was first performed for 10 min at 25 °C, and then at 42 °C for one hour. The RT reaction was stopped by incubating the samples for 10 min at 70 °C. The qPCR was performed with 1× iTaq SYBR Green Supermix (Biorad), 0.4 µM of each primer, and 1 µl of the cDNA sample. mRNA levels were determined by qPCR using sequence-specific primers for the indicated transcripts. qPCR primers designed using Primer-BLAST (NCBI) are listed in Supplementary Table 2. Normalized transcript expression ratios from three independent experiments were determined using the Livak method[72].

### Polysome profiling

Polysome profiles were performed as described in Kuzuoglu-Ozturk et al.[73]. HEK293T cells were pre-treated with cycloheximide (50 µg/ml) for 30 min. Lysates were prepared in lysis buffer (10 mM Tris-HCl pH = 7.4, 10 mM NaCl, 1.5 mM MgCl$_2$, 0.5% Triton X-100, 2 mM DTT, 50 µg/ml cycloheximide) and polysomes separated on a 10–50% sucrose gradient in gradient buffer (10 mM Tris-HCl pH 7.4, 75 mM KCl, 1.5 mM MgCl$_2$). Polysome fractions were collected using the Teledyne Isco Density Gradient Fractionation System.

To isolate RNA from sucrose fractions, samples were first digested with proteinase K (Sigma Aldrich, 1% of the sample volume; 100 mg/ml in 50 mM Tris-HCl pH = 8.1, 10 mM CaCl$_2$ buffer) at 37 °C for 45 min and shaking at 400 rpm. The digested samples were mixed with 1 volume of Phenol:Chloroform:Isoamyl alcohol (PanReac AppliChem, 25:24:1, v/v), vortexed, and spun down 5 min at 20,000 × *g* at 4 °C.

Supernatants were transferred into three volumes of 100% ethanol, 0.1 volumes of 3 M NaOAc pH = 5.2 and 1 µl of GlycoBlue, and precipitated at −20 °C. Samples were pelleted for 30 min at 20,000 × *g* and 4 °C, washed once with 100% ethanol and another time with 70% ethanol, dried, and resuspended in 30 µl H$_2$O. Fractions were reverse transcribed and analyzed by qPCR.

### RNA pulldown

For the RNA pulldown, 3 × 10$^6$ HEK293T cells were plated in 10 cm plates and transfected using Lipofectamine 2000 (Invitrogen) with the following plasmids expressing V5-SBP fusions: MBP (1.5 µg), DAP5 FL (4 µg) and eIF4A* (15 µg), MIF4G (4 µg) or ΔW2 (6 µg) mutants, eIF4G FL (15 µg) or ΔN (4 µg), and Chimeras (4 µg). A detailed description of the RNA pulldown procedure can be found in Kuzuoglu-Ozturk et al.[73]. Cells were harvested 48 hours post transfection, washed with ice-cold PBS, and lysed on ice for 15 min in 500 µl of NET buffer [50 mM Tris-HCl pH = 7.5, 150 mM NaCl, 0.1% Triton X-100, 1 mM EDTA pH = 8.0, 10% glycerol, supplemented with 1× protease inhibitors (Roche)]. Cell debris was removed by centrifugation at 16,000×*g* and 4 °C. Input samples (5% of the total) were collected for western blotting and RT-qPCR. Cell lysates were immediately incubated with 50 µl of a 50% slurry of streptavidin beads pre-incubated with yeast RNA (250 µg of yeast RNA/100 µl of 50% slurry). Beads were washed three times with NET buffer and resuspended in 1 ml of NET buffer without detergent. An aliquot (20% of the total) of the bead suspension was mixed with SDS-PAGE sample buffer for western blotting after centrifugation to pellet the resin. The remaining beads were used for RNA isolation with TriFast (Peqlab Biotechnologies). cDNA of the input and precipitated fractions (20% each) was prepared and analyzed using qPCR (5% of the cDNA), as described above. The list of primers used for the qPCR experiments can be found in Supplementary Table 2.

### Pulldown assays

Pulldown assays were performed in the presence of RNase A. HEK293T cells were grown in 10 cm dishes and transfected using Lipofectamine 2000 (Invitrogen) according to the manufacturer's recommendations. The transfection mixtures in Supplementary Fig. 5a, b contained 1.5 µg of V5-SBP-MBP, 4 µg of V5-SBP-DAP5, and 5 µg of GFP-eIF2β. After transfection, cells were treated as described in the RNA-pulldown section, with the exception that the streptavidin beads were not incubated with yeast RNA and the samples were solely used for immunoblotting.

For the cap pulldown, the transfection mixtures contained 1 µg GFP-MBP or 12 µg GFP- chimeric-4EBP. Cap-bound proteins were pulled down using γ-Aminophenyl-m$^7$GTP beads (Jena Bioscience).

### Flow cytometry

Cells were seeded (0.6 × 10$^6$ WT and 0.7 × 10$^6$ DAP5-null HEK293T cells) in six-well plates 24 hours before transfection. Transfections were carried out with Lipofectamine 2000 (Invitrogen), with the following transfection mixtures: WNK1-mNG2$_{11}$-EBFP (0.35 µg), mNG2$_{1-10}$ (1 µg), mCherry (10 ng), V5-SBP-MBP (0.25 µg) or DAP5 (0.65 µg). 48 hours after transfection, cells were trypsinized, sedimented (1000 rpm for 3 min at room temperature), resuspended in 1% FBS in PBS, and analyzed using the Becton Dickinson FACSMelody™ Cell Sorter and FlowJo software (Becton Dickison). To determine mNG2, EBFP, and mCherry-positive events, we analyzed non-transfected and control transfected cells. Cut-offs were applied uniformly for all measured conditions.

### Knockdowns

0.64 × 10$^6$ HEK293T cells were transfected with 2 µg pSUPERpuro scramble control or ABCE1 shRNAs, after seeding in six-well plates, using Lipofectamine 2000 (Invitrogen). 24 hours after transfection cells were treated with 3 µg/ml puromycin (Serva Electrophoresis) for 24 h. Selected cells were re-seeded and re-transfected with DNA

mixtures containing 0.5 µg of *WNK1*-R-LUC reporter plasmid and 0.25 µg of the control GFP-F-LUC reporter.

## Statistics and reproducibility

The manuscript contains only reproducible experiments and data. Each experiment was performed in three biological replicates, with the exception of the RNA-Seq and Ribo-Seq data sets where only two biological replicates were used. All the replications were successful.

Figure 1b, Supplementary Fig. 1d. Upregulated and downregulated genes were identified using $\log_2$Fold Change (FC) between null and control cells >0 or <0, respectively, and FDR < 0.005.

Figure 1c. The quantitative value represented in the graphs corresponds to $-\log_{10}(q$ value) determined by the GOseq analysis tool[65].

Figure 1h, i. Boxes indicate the 25th to 75th percentiles; black line inside the box represents the median; whiskers indicate the extend of the highest and lowest observations; dots show the outliers.

Figures 6j, k. Boxes represent the 25th to 75th percentiles; black line shows the median and the cross the average; whiskers show the variability outside the upper and lower quartiles; dots show the outliers. Significance was determined by one-sided Wilcoxon rank-sum test and indicated if $p < 2.2e^{-16}$ (**).

Figure 1j. The quantitative values represented in the pie chart indicate the percentage of uORFs containing canonical and near-cognate start codons or other codon sequence in the 5′ leaders of the DAP5 targets.

Figures 2–5, 7, Supplementary Figs. 1k–m, 4e–g, 5f, g, j–o, 6, 7. The quantitative value that is graphed represents the mean mRNA, protein level, or cell values; error bars represent standard deviations from three independent experiments. All values were calculated using Microsoft Excel statistical tools. Significance was determined using the one-way ANOVA test. In the RT-qPCR experiments, normalized transcript expression ratios from three independent experiments were determined using the Livak method[72].

Supplementary Fig. 4a–d. Length, GC content, minimum free energy, and TE were determined for the 5′ leaders of DAP5 targets and in all other mRNAs expressed in HEK293T cells. Statistical significance was calculated with the one-sided Wilcoxon rank-sum test.

Supplementary Figs. 5c and 7n. The hypergeometric test (phyper) in R was applied to estimate the likelihood of list overlap.

Supplementary Data 1. The statistical significances calculated by RiboDiff[28] were based on a generalized linear model.

## Reporting summary

Further information on research design is available in the Nature Portfolio Reporting Summary linked to this article.

## Data availability

The datasets generated in this study have been deposited in NCBI's Gene Expression Omnibus and are accessible through GEO Series accession number GSE155854. Imaging data are available at Mendeley Data with the https://doi.org/10.17632/bzpfcnzg8w.1. Harringtonine, LTM, and QTI datasets from human HEK293 cells were downloaded from the Sequence Read Archive database with the accession numbers SRA056377 and SRA160745. RocA and DENR datasets were retrieved from the GEO database with the accession numbers GSE70211 and GSE140084, respectively. Further information, resources, and reagents are available from the corresponding author(s) upon reasonable request. Source data are provided with this paper.

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

## Acknowledgements

We dedicate this work to the memory of Elisa Izaurralde and acknowledge that the study was conceived and carried out in her laboratory. We are thankful to Markus Landthaler and Ulrike Zinnall for their help with ribosome profiling, and Heike Budde for cloning the pcDNA3.1-MCS-mCherry plasmid. This work was supported by the Max Planck Society.

## Author contributions

R.W. designed and conducted the experiments assisted by M.-Y.C. L.K. performed luciferase assays and generated several constructs. I.H. assisted and contributed to the analysis of FACS data. E.V. contributed to data analysis. R.W. and C.I. conceived the project, interpreted the results, and wrote the manuscript. All authors read and corrected the manuscript.

## Funding

## Competing interests

The authors declare no competing interests.
