## [Peer Review File · Nature Communications]

DAP5 enables main ORF translation on mRNAs with structured and uORF-containing 5' leadersREVIEWER COMMENTS

Reviewer #1 (Remarks to the Author):

This is a very nice story identifying DAP5 as a novel component required for translation re-initiation on certain mRNAs. Almost half of all mRNAs in humans have upstream ORFs (uORFs) which are often translated. Understanding how ribosomes are able to translate a uORF and then re-initiate on the main CDS downstream is an exciting topic of interest at the moment. Many important genes have uORFs, making this of relevance for both normal development as well as disease. Hence the identification of a new factor involved in this process will be of interest to a wide audience. Furthermore, the data are of good quality and quite comprehensive. I have only one major issue which would be good to address prior to publication:

Major Issue:

I think it is clear that DAP5 is involved in reinitiation, and not leaky scanning, from a number of experiments shown in the manuscript. For instance, there are 80S footprints on the WNK1 uORFs (indicating that they are indeed translated), extension of the uORFs reduces expression from the main CDS (which fits with the fact that ribosomes are incompetent to re-initiate translation after long uORFs), etc.

That said, it does not look like DAP5 is always required for re-initiation (ie on all mRNAs undergoing reinitiation) as can be seen from the only-partial overlap with targets of other re-initiation factors (Suppl. Fig. 7e). Hence DAP5 must be required for reinitiation on a subset of mRNAs, and it is unclear what feature these mRNAs need to possess to be DAP5 dependent. This would broaden our understanding of DAP5 function and requirement beyond the specific example of WNK1. I think there are some experiments that are not too complicated, and hence worth trying, to achieve this broader understanding.

The manuscript suggests there is a requirement for secondary structure in the 5'UTR of DAP5 targets. Furthermore, the manuscript presents the surprising and counter-intuitive result that a 5' deletion (delta 2) of the 5' end of the WNK1 R-LUC reporter completely shuts down translation of the reporter (Fig. 3f). Normally, one would expect that removing uORFs from a 5'UTR will lead to stronger translation of the main CDS, but the opposite is happening here. Hence this result, as it currently stands in the manuscript, is a bit mysterious and not understood. This result strongly suggests, however, that there is indeed an inhibitory element in what is left of the WNK1-delta2 reporter 5'UTR, and this might be the secondary structure the authors are looking for. I can think of a couple possible explanations: 1) Maybe there is a hairpin at the new 5' end of the WNK1-delta2 reporter (ie directly downstream of the delta-2 deletion breakpoint) that is normally internal to the 5'UTR but is now directly adjacent to the 5'cap. Such a structure is known to be particularly bad close to 5'caps because it inhibits PIC recruitment to the mRNA. 2) Alternatively, there might be some secondary structure directly after the delta-2 breakpoint that is strongly inhibitory regardless of its position relative to the 5'cap, and normally translation of the uORFs upstream of this secondary structure (in the delta-2 deleted region) helps to open it up, allowing ribosomes to get through. If the upstream region is deleted, this doesn't happen and the secondary structure becomes strongly inhibitory. Either way, the delta-2 reporter must have a strongly inhibitory element that is not inhibitory in the context of the full-length WNK1 reporter. It seems that replacement of eIF4G with DAP5 is required to get past this inhibitory element. I would guess this inhibitory element is dictating the DAP5 requirement on an mRNA. This could be teased apart as follows:

1. Does mutation of all 8 uORF start codons in the WNK1 reporter yield a reporter that is well-expressed and DAP5 independent, or a reporter that is repressed and DAP5 independent? If the reporter is constitutively repressed, it would help prove that there is an inhibitory activity within the 5'UTR in addition to the uORFs, and would suggest explanation #2 is the correct one.

Furthermore, either result would help prove the author's claim that "initiation at the main AUG only occurs after uORF translation."

I think in its current state, the manuscript does not yet convincingly show this because the experiments extending the uORFs to become longer simply shows that long uORFs are inhibitory for re-initiation, which is already known.

2. If the uORF start codon mutation yields a reporter that is constitutively repressed, then nested 5' truncations of this reporter should at some point remove this inhibitory element, yielding a well-translated reporter. This will identify the element. Is it a hairpin?

3. If instead of deleting the delta-2 region, it is replaced with some generic sequence lacking uORFs, does this also lead to a strongly repressed reporter or not? This will distinguish whether the repression of the delta-2 reporter comes from proximity of this inhibitory element to the 5'cap, or whether it comes from removal of the uORFs.

I think these three experiments with luciferase reporters will delineate the identity of this inhibitory element that is required for making an mRNA DAP5 dependent.

Minor Issues:

1. The manuscript claims

"DAP5-dependent translation of R-LUC was abolished in the reporters with the long uORF2s (Fig. 4b, c)."

To me it looks like the reporter is still DAP5 dependent (the delta stop1 reporter still goes down in the DAP5 null and back up again upon DAP5 re-expression, Fig 4b) however the overall level of expression of the reporter is much lower, likely due to the long uORF which is known to not allow re-initiation. In fact, this interpretation fits with the length series shown in Suppl. Fig. 5b where the longer uORFs are still DAP5 dependent, just that the overall expression level is progressively lower.

The fact that longer uORFs are inhibitory for translation of the main CDS is well established, hence the uORF extensions show in Fig 5 and Suppl. Fig. 5 are nice supporting evidence that these uORFs are indeed translated, but it doesn't show that the uORF translation is needed for translation of the main CDS. It just shows that introducing a long inhibitory uORF blocks main CDS translation, which is expected. In sum, I would not conclude from this that it turns the reporter DAP5 independent. It simply becomes difficult to see the DAP5 dependence because a strongly inhibitory element is introduced into the 5'UTR.

The fact that uORF translation is required for DAP5 dependence would be shown by the experiment suggested in Main Issues #1 above: mutating all uORF start codons.

In my opinion, the DAP5 dependence is coming from another element, and the uORFs are needed to swap eIF4G for DAP5.

2. It would be helpful to show the molecular characterization of the DAP5 alleles that are described in the materials & methods (the 172bp deletion in exon 10, etc..) as a supplementary figure so it is more visually obvious.

3. Figure 1d – it would be good to complement this figure panel with a Q-RT-PCR for mRNA levels of these targets (ROCK1, WNK1, INPPL1) to show that what is shown here is a translational effect (as expected from the RNA-seq analysis that was done).

4. I find it somewhat confusing in the legend of Fig 1e and in the Results text to write that the uORFs shown in green are 'in frame' with the main CDS, because there are stop codons separating these

uORFs from the main CDS. Perhaps this can be clarified by writing 'in the same reading frame' rather than 'in frame' which would suggest they are fused together to the main CDS.

5. Page 6, the authors write "Moreover, the uORFs translated in the null cells concentrated in the regions of the 5' leaders adjacent to high propensity for structure (Fig. 1e, Supplementary Fig. 2)." I don't see how this is visible in the referenced figures?

6. Fig 2a-c: the MA3 chimera seems to rescue pretty well... This could perhaps be commented and discussed.

7. The authors write "Expression of the mNG2 plasmids in trans did not generate a yellow-green signal (Supplementary Fig. 6a)."

It's not clear what the authors mean by 'in trans' ? Do they mean separately in two different cell populations? As a geneticist, 'in trans' usually means on separate chromosomes, or separate plasmids, within the same cell or organism. I guess this is not what the authors mean here?

Reviewer #2 (Remarks to the Author):

In this work, authors show that the non-canonical initiation factor death-associated protein 5 (DAP5) is involved in translation re-initiation in mRNAs with long, structure-prone 5' leader sequences and persistent upstream open reading frame (uORF) translation. Mechanistically, they show that cap/eIF4F- and eIF4A-dependent recruitment of DAP5 to the mRNA facilitates re-initiation by unrecycled post-termination 40S subunits.

The manuscript is an interesting piece of work. It is well-written, logically presented, and includes the appropriate controls. However, there are a few points that should be clarified.

Specific comments:

In this work, authors focused their work in transcripts, like WNK1, ROCK1 and AKT1, carrying several uORFs. However, the impact of the different uORFs on translational control of these transcripts is not shown. It is not clear which uORFs are indeed functionally relevant for translational regulation of the main ORF. This point should be clarified as it might be important to understand the data presented.

Fig. 1: Authors say « Close inspection of the RFP profiles revealed that cap-proximal uORF translation is DAP5-independent whereas downstream uORFs and CDS are translated in a DAP5-dependent manner (Fig. 1e, f) .» This conclusion is from ribosome profiling data. It would be interesting to complement these data by assessing the potential function of DAP5 in the translation of WNK1 downstream uORFs, using a different method.

Fig. 5: It is not clear if the constructs presented in Fig. 5a carry uORFs 1 to 5. If the constructs carry uORFs 1 to 5 upstream of uORF6, how the authors can be sure that the function of DAP5 in translation re-initiation is exclusively for the main ORF? Can it be operating in translation re-initiation at uORF6 after translation of an upstream uORF?

Reviewer #3 (Remarks to the Author):

In this manuscript, Weber et al. (2021) interrogate the role of DAP5 in translation initiation. DAP5 (otherwise known as eIF4G2) is a non-canonical initiation factor which is homologous to the c-terminal two-thirds of the canonical eukaryotic initiation factor 4G1 (eIF4G1). To date, a mechanistic role for

DAP5 has remained elusive. Proposed roles include its involvement in internal ribosome entry site (IRES)-dependent translation and interactions with non-canonical cap-binding complexes.

The authors begin this study by performing ribosome footprint profiling on wild-type and DAP5-null cells. Through this analysis, the authors find that DAP5 is important for main ORF (mORF) translation of a subset of mRNAs characterized by long, structured, uORF-containing 5'-leaders. Subsequent mutational analysis with renilla luciferase (RLuc) reporters reveals that all characterized domains of DAP5 are essential to its function and are non-exchangeable with the analogous domains in eIF4G1. Overexpression of a 4E binding protein (4EBP) construct prevents DAP5-dependent translation, suggesting that DAP5 associates with capped mRNAs and is dependent on the eIF4E-eIF4G interaction for its function. This is particularly interesting considering that DAP5 does not have an eIF4E binding domain. It would have been a welcome addition for the authors to further explore how its function relies on the eIF4E-G interaction. This takes us through the beginning of Figure 3.

Despite these insights into DAP5 function, the remainder of the paper is plagued by inconclusive experiments which are difficult to interpret in the context of the authors' major claim: that is, that DAP5 is essential for translation re-initiation on uORF-containing mRNAs. The alternative hypothesis which goes unmentioned and untested is that DAP5 could promote leaky scanning. For the remainder of the paper, the authors perform many RLuc reporter experiments in which they modify the 5'UTR of (primarily) WNK1 (a DAP5-dependent gene) and find that these modifications results in a loss of DAP5-dependent mORF translation (Figures 3-7). While these experiments might test DAP dependent translation of the mORF, they are not set up to test their hypotheses and are completely confounded by the complexity of the WNK1 5'UTR which contains 8 different uORFs. What would be required to test the alternative hypotheses of reinitiation and leaky scanning would be a simpler 5' UTR with a single uORF at play. A good starting experiment would be to mutate the start codon and show that uORF mediated regulation is lost and that mORF translation becomes DAP5 independent. Mutation of the stop codons in the complicated WNK1 5'UTR may test something related to the importance of uORF length in regulation of mORF translation, but these ideas are not particularly novel (REF). In light of these many limitations, the major claim of the paper that DAP5 promotes translation re-initiation is unsupported by the experiments provided.

Major points:

1. While it is standard in the field to use drugs such as harringtonine and lactidomycin to identify translation initiation sites, these experiments suffer from a likely overestimation of uORF utilization. What happens in cells treated with these drugs is that all ribosomes run off of all the mRNAs in the cell, thus dramatically increasing the concentration of ribosomes in the cell. In this situation, it seems likely that ribosomes will fill first all the AUG start codons in the cell (and become trapped) and then they will move on to less optimal initiation sequences. This likely explains the very abundant use of non-canonical start codon usage revealed in these experiments. The methods developed by Shu-bin Qian where ribosomes are trapped with drugs at start sites and cleared in lysate by puromycin likely do a better job of capturing the authentic biology of translation initiation.
2. It would be useful to use some reporters with uORFs that do not respond to DAP5 as negative controls. While it is nice to see three that emerged as strong candidates from ribosome profiling, it should be easy to find some candidates that share the feature of having a 5' uORF but which behave differently to DAP5.
3. Throughout the manuscript, no statistical tests are performed. For example in Figure 4b, p-values would be helpful for the Δ STOP constructs. It is difficult for the reader to reasonably interpret whether the conclusions are correct without statistics to back up the data.
4. In Fig 3D, the authors argue that because a large deletion of the 5'UTR (Δ 2) abolishes RLuc expression, uORFs must be mandatory for translation of the mORF. This experiment just doesn't make

sense. First, it goes against a large body of literature suggesting that uORFs are usually inhibitory to translation of the downstream mORF (see PMID: 19372376 for one example). Second, the experiment is set up incorrectly. To determine the importance of uORF translation to mORF translation, the uORF start codons should simply be mutated (i.e. AUG \diamond AAA). Additionally, the fact that the $\Delta 1$ mutation causes mORF expression to go up instead of down argues against the author's general claim that uORF translation is "mandatory" to downstream mORF translation because in this case, deletion of uORFs in this region leads to increased translation from the mORF (lane 3 vs. lane 6).

5. In Figures 4, 5, and 6 the authors claim that lengthening the respective uORFs causes WNK1 reporter translation to no longer be dependent on DAP5. However, in all three cases, lengthening the uORF completely abrogates the mORF signal (up to ~ 100 -fold decrease in some cases) – but even at these low levels of expression there appears to remain some DAP5 dependence. Given the low signal for this assay, it could be important to do a control experiment to show that the assay is in the dynamic range.

6. In Figures 4C and 5C the FLuc expression is consistently decreased in every single DAP5-null line. Presumably, the FLuc construct should be a transfection control and should not encode a DAP5-dependent 5'UTR or uORFs. The observation that FLuc levels are consistently responding to DAP5 knockout is troublesome.

7. The conclusions from Figure 5D may be misinterpreted. The fact that DAP5-KO causes N-terminally-extended RLuc expression to increase and mORF expression to decrease (compare lanes 3 and 4) could support a leaky scanning model (i.e. if loss of DAP5 prevents scanning past these extended uORFs).

8. The experiments in Figure 7 show modest effects and it is not clear how they support the overall conclusions of the manuscript. Perturbation of termination/recycling could certainly have an impact on re-initiation but the data are just not very compelling.

Overall, the paper relies too heavily on reporter mRNAs and mutational analyses, without directly assessing the role of the ribosome in translating the uORFs. I appreciate that Figure 6 attempted to address this point, but I think the stronger experiment is to delete the start site (or rather all 8 of them) rather than to remove stop codons. The reporter is too complicated and it is impossible to know what is really happening to ribosomes as they traverse the 5' UTR of the gene, and therefore how this might impact downstream mORF expression. As such, while it is clear that DAP5 is an important protein and understanding its non-canonical roles in translation initiation is of value, the manuscript fundamentally fails to clarify DAP5's function within the cell.

NCOMMS-21-06391A

We would very much like to thank the referees for taking the time to review our work so carefully and constructively. We are grateful for all their input and believe that addressing their comments and suggestions have improved our revised manuscript greatly.

Please find below an outline of the changes made to the manuscript (in **blue**) to address the specific comments raised (in black).

Reviewer 1

This is a very nice story identifying DAP5 as a novel component required for translation re-initiation on certain mRNAs. Many important genes have uORFs, making this of relevance for both normal development as well as disease. Hence the identification of a new factor involved in this process will be of interest to a wide audience. Furthermore, the data are of good quality and quite comprehensive.

We thank the reviewer for their positive comments.

I have only one major issue which would be good to address prior to publication. I think it is clear that DAP5 is involved in reinitiation, and not leaky scanning, from a number of experiments shown in the manuscript. That said, it does not look like DAP5 is always required for re-initiation (ie on all mRNAs undergoing reinitiation) as can be seen from the only-partial overlap with targets of other re-initiation factors. Hence DAP5 must be required for reinitiation on a subset of mRNAs, and it is unclear what feature these mRNAs need to possess to be DAP5 dependent. This would broaden our understanding of DAP5 function and requirement beyond the specific example of WNK1. I think there are some experiments that are not too complicated, and hence worth trying, to achieve this broader understanding.

We agree with the reviewer that DAP5 is required for re-initiation of translation in only a group of mRNAs characterized by structured and uORF-rich 5' leaders. Accordingly, we have highlighted this conclusion in the text of the revised manuscript.

The manuscript suggests there is a requirement for secondary structure in the 5'UTR of DAP5 targets. Furthermore, the manuscript presents the surprising and counter-intuitive result that a 5' deletion (delta 2) of the 5' end of the WNK1 R-LUC reporter completely shuts down translation of the reporter (Fig. 3f). Normally, one would expect that removing uORFs from a 5'UTR will lead to stronger translation of the main CDS, but the opposite is happening here. Hence this result, as it currently stands in the manuscript, is a bit mysterious and not understood. This result strongly suggests, however, that there is indeed an inhibitory element in what is left of the WNK1-delta2 reporter 5'UTR, and this might be the secondary structure the authors are looking for. I can think of a couple possible explanations:

1) Maybe there is a hairpin at the new 5' end of the WNK1-delta2 reporter (i.e. directly downstream of the delta-2 deletion breakpoint) that is normally internal to the 5'UTR but is now directly adjacent to the 5'cap. Such a structure is known to be particularly bad close to 5'caps because it inhibits PIC recruitment to the mRNA.

2) Alternatively, there might be some secondary structure directly after the delta-2 breakpoint that is strongly inhibitory regardless of its position relative to the 5'cap, and normally translation of the uORFs upstream of this secondary structure (in the delta-2 deleted region) helps to open it up, allowing ribosomes to get through. If the upstream region is deleted, this doesn't happen and the secondary structure becomes strongly inhibitory. Either way, the delta-2 reporter must have a strongly inhibitory element that is not inhibitory in the context of the full-length WNK1 reporter. It seems that replacement of eIF4G with DAP5 is required to get past this inhibitory element. I would guess this inhibitory element is dictating the DAP5 requirement on an mRNA.

We appreciate the scenarios discussed by the reviewer and agree with the interpretation that strong and structured inhibitory elements in the 5' leader targets dictate the requirement for DAP5 in the translation of a selected group of mRNAs. To explain the absence of main CDS translation in the WNK1-Δ2 reporter we provide in the revised manuscript additional experiments. We observed that the *WNK1* (and most likely other DAP5 targets) 5' leader contains a potential G-quadruplex (Gq) motif. The cap-proximal truncation of the 5' leader (Δ2) leaves the Gq motif adjacent to the cap. In agreement with the reviewer's interpretation that strong secondary structures next to the cap prevent PIC recruitment, addition of an unstructured sequence of 18 CAA repeats to the truncated mRNA restored DAP5-dependent translation of R-LUC (see Fig. 3e-j). Furthermore, deletion of the Gq motif or its substitution with the CAA repeats sustained R-LUC translation in a DAP5-independent manner (see Fig. 3e-j). Thus, DAP5 targets contain highly structured RNA elements that modulate main CDS translation and impose the requirement of DAP5, instead of eIF4G.

This could be teased apart as follows:

1. Does mutation of all 8 uORF start codons in the WNK1 reporter yield a reporter that is well-expressed and DAP5 independent, or a reporter that is repressed and DAP5 independent? If the reporter is constitutively repressed, it would help prove that there is an inhibitory activity

within the 5'UTR in addition to the uORFs, and would suggest explanation #2 is the correct one.

We have performed the experiment requested by the reviewer and altered all cognate and near-cognate start codons that showed 80S footprints in the *WNKI* 5' leader (Δ START reporter, Fig. 5). We observed that reduction of uORF translation resulted in a 1.6x-fold increase in R-LUC translation, indicating that uORF translation limits main CDS translation (Fig. 5). Additionally, R-LUC remained DAP5-dependent. We conclude that as long as the inhibitory activity of the structured elements is present in the *WNKI* 5' leader, DAP5 is crucial for initiation of translation at the main CDS.

Although we have suppressed the initiation of translation at the start sites displaying 80S footprints in the Ribo-Seq data, the *WNKI* 5' leader harbours 47 potential cognate and near-cognates start codons (4 AUG, 19 CUG, 4 UUG and 20 GUG, 12 upstream of the Gq motif) which can be recognized by the PICs stalled or slowed-down by the strong structured elements and used to initiate translation. Moreover, 22% of uORF translation in DAP5 targets can initiate at codons other than AUG or near-cognate (Fig. 1j). Thus, uORF translation can still occur in the Δ START reporter. To further interfere with uORF translation, we removed the termination codons of all reading frames in the *WNKI* 5' leader (Δ STOP reporter, Fig. 5). In the absence of translation termination at the 5' leader, only an N-terminally extended version of R-LUC was produced and independently of DAP5 (Fig. 5). The presence of a long R-LUC indicates that initiation of translation occurred in the 5' leader upon the recognition of upstream initiation codons by PICs stalled or slowed-down by the strong inhibitory features of the structured RNA elements. The translating ribosomes unwind the secondary structure and continue to translate until they find a STOP codon. If the ribosome translates in the same reading frame of the main CDS, the resulting product is an N-terminally extended R-LUC. In this context, DAP5 is not required for translation initiation. We have also generated a *WNKI* Δ START+STOP reporter. None of the R-LUC proteins was synthesized. These results suggest that DAP5-dependent initiation at the main AUG requires a preceding termination event, i.e., DAP5 promotes re-initiation after uORF translation.

2. If the uORF start codon mutation yields a reporter that is constitutively repressed, then nested 5' truncations of this reporter should at some point remove this inhibitory element, yielding a well-translated reporter. This will identify the element. Is it a hairpin?

As stated above, *WNK1* (and most likely other DAP5 targets) 5' leader contains a strong secondary structure (predicted G-quadruplex motif). As expected by the reviewer, deletion of this motif from *WNK1* 5' leader increased R-LUC translation efficiency (~2.5x-fold) (Fig. 3). Thus, strong structured RNA elements such as Gq motifs in the 5' leaders of DAP5 targets have inhibitory effects on main CDS translation which can be overcome by DAP5.

3. If instead of deleting the delta-2 region, it is replaced with some generic sequence lacking uORFs, does this also lead to a strongly repressed reporter or not? This will distinguish whether the repression of the delta-2 reporter comes from proximity of this inhibitory element to the 5' cap, or whether it comes from removal of the uORFs.

As suggested by the reviewer, addition of an unstructured sequence of 18 CAA repeats to the *WNK1* $\Delta 2$ truncated mRNA restored DAP5-dependent translation of R-LUC indicating that the proximity of the structured inhibitory element to the 5' cap prevents PIC recruitment and/or function (Fig. 3).

Minor issues:

1. The manuscript claims “DAP5-dependent translation of R-LUC was abolished in the reporters with the long uORF2s (Fig. 4b, c).” To me it looks like the reporter is still DAP5 dependent (the delta stop1 reporter still goes down in the DAP5 null and back up again upon DAP5 re-expression, Fig 4b) however the overall level of expression of the reporter is much lower, likely due to the long uORF which is known to not allow re-initiation. In fact, this interpretation fits with the length series shown in Suppl. Fig. 5b where the longer uORFs are still DAP5 dependent, just that the overall expression level is progressively lower.

We agree with the reviewer and have rephrased the text accordingly.

The fact that longer uORFs are inhibitory for translation of the main CDS is well established, hence the uORF extensions show in Fig 5 and Suppl. Fig. 5 are nice supporting evidence that these uORFs are indeed translated, but it doesn't show that the uORF translation is needed for translation of the main CDS. It just shows that introducing a long inhibitory uORF blocks main CDS translation, which is expected. In sum, I would not conclude from this that it turns the

reporter DAP5 independent. It simply becomes difficult to see the DAP5 dependence because a strongly inhibitory element is introduced into the 5'UTR. In my opinion, the DAP5 dependence is coming from another element, and the uORFs are needed to swap eIF4G for DAP5.

We agree with the reviewer and have rephrased the text accordingly.

2. It would be helpful to show the molecular characterization of the DAP5 alleles that are described in the materials & methods (the 172bp deletion in exon 10, etc..) as a supplementary figure so it is more visually obvious.

We have followed the reviewer's suggestion and included in Supplementary Fig. 3 the trace peaks and sequencing results of DAP5 genomic region in wild type and DAP5-null cells.

3. Figure 1d – it would be good to complement this figure panel with a Q-RT-PCR for mRNA levels of these targets (ROCK1, WNK1, INPPL1) to show that what is shown here is a translational effect (as expected from the RNA-seq analysis that was done).

We have quantified *ROCK1*, *WNK1* and *INPPL1* mRNA levels using qPCR, as suggested by the reviewer. The results are included in Supplementary Fig. 1g.

4. I find it somewhat confusing in the legend of Fig 1e and in the Results text to write that the uORFs shown in green are 'in frame' with the main CDS, because there are stop codons separating these uORFs from the main CDS. Perhaps this can be clarified by writing 'in the same reading frame' rather than 'in frame' which would suggest they are fused together to the main CDS.

We agree with the reviewer and have rephrased the text accordingly.

5. Page 6, the authors write "Moreover, the uORFs translated in the null cells concentrated in the regions of the 5' leaders adjacent to high propensity for structure (Fig. 1e, Supplementary Fig. 2)." I don't see how this is visible in the referenced figures?

To clarify our conclusions, we have rephrased this sentence in the revised manuscript. We now indicate that the density of RFPs in DAP5-target mRNAs decreased following the predicted structured region in the 5' leaders of each transcript (page 7).

6. Fig 2a-c: the MA3 chimera seems to rescue pretty well... This could perhaps be commented and discussed.

We agree with the reviewer and have rephrased the text accordingly.

7. The authors write “Expression of the mNG2 plasmids in trans did not generate a yellow-green signal (Supplementary Fig. 6a).” It’s not clear what the authors mean by ‘in trans’? Do they mean separately in two different cell populations? As a geneticist, ‘in trans’ usually means on separate chromosomes, or separate plasmids, within the same cell or organism. I guess this is not what the authors mean here?

To clarify the text, we have rephrased the sentence.

Reviewer 2

The manuscript is an interesting piece of work. It is well-written, logically presented, and includes the appropriate controls.

We thank the reviewer for their positive criticism.

In this work, authors focused their work in transcripts, like *WNK1*, *ROCK1* and *AKT1*, carrying several uORFs. However, the impact of the different uORFs on translational control of these transcripts is not shown. It is not clear which uORFs are indeed functionally relevant for translational regulation of the main ORF. This point should be clarified as it might be important to understand the data presented.

The data described in the revised manuscript indicates that DAP5 is required for initiation of translation at the main CDS in transcripts with highly structured 5' leaders and multiple uORFs. The intricate nature of these 5' leaders interferes with scanning of PICs which tend to recognize and translate uORFs even under suboptimal sequence contexts. The Ribo-Seq data shows that cap-proximal uORFs are frequently translated in DAP5 targets, most likely because the scanning complexes have difficulties to move past the strong and repressive structured elements. In our attempts to manipulate uORF features we observed that uORF length, but not uORF identity or location, regulates initiation of translation at the main CDS (Fig. 4 and Supplementary Fig. 6). However, uORF translation is not sufficient to mediate DAP5-dependent initiation of translation. Deletion of the strong structured elements in the 5' leader of the *WNK1* reporter was sufficient to induce DAP5-independent translation of R-LUC. Our results support a model where strong secondary structure elements elicit pervasive uORF translation and limit main CDS translation. The inhibitory role of these elements is then bypassed by DAP5. Together with eIF4A, DAP5 fuels ribosomal complexes that translated uORFs to move through the structured regions of the 5' leader and initiate translation at a new start codon.

Fig. 1: Authors say « Close inspection of the RFP profiles revealed that cap-proximal uORF translation is DAP5-independent whereas downstream uORFs and CDS are translated in a DAP5-dependent manner (Fig. 1e, f).» This conclusion is from ribosome profiling data. It

would be interesting to complement these data by assessing the potential function of DAP5 in the translation of WNK1 downstream uORFs, using a different method.

We have followed the suggestion of the reviewer and tried to monitor translation initiation at the uAUG of uORF6 which shows reduced RFPs in DAP5-null cells. We have generated a *WNK1*-R-LUC reporter where all termination codons between uORF6 and R-LUC were altered and uORF6 is in the same reading frame as the main CDS. The idea was to detect the synthesis of an N-terminally extended R-LUC in wild type and null cells using western blot. However, we observed that due to the intricate nature of the 5' leader, initiation translation occurs frequently at the cap-proximal uORFs located before the highly structured region but not at the uORFs (such as uORF6) downstream of this region. Thus, the R-LUC protein resulting from initiation at uORF6 AUG was not detected by western blot (only an extended version of R-LUC with a translation start site at one of the cap-proximal uORFs) (see Fig. I included in the point-by-point answer). We were therefore unable to clarify the reviewer's question. We hope that in the future we are able to address this question using distinct methodologies, such as 40S profiling in wild type and DAP5-null cells.

Fig. I. WT and DAP5-null cells were transfected with plasmids expressing *WNK1*-R-LUC reporters (a), V5-SBP-MBP or V5-SBP-DAP5 and F-LUC-GFP. The immunoblot in b shows the expression levels of the proteins used in the assay. Membranes were incubated with anti-V5, GFP, and R-LUC antibodies. Notice the two forms of R-LUC synthesised upon initiation at the main AUG (approximately 35 kDa) or at a cap proximal uORF start codon (close to 70 kDa).

Fig. 5: It is not clear if the constructs presented in Fig. 5a carry uORFs 1 to 5. If the constructs carry uORFs 1 to 5 upstream of uORF6, how the authors can be sure that the function of DAP5 in translation re-initiation is exclusively for the main ORF? Can it be operating in translation re-initiation at uORF6 after translation of an upstream uORF?

The constructs presented in Fig. 5a (Fig. 4a in the revised manuscript) still carry all the other uORFs present in the 5' leader of *WNK1*. The only change in the reporter is the modification of the termination codons downstream of the uORF6 AUG. We have included a sentence in the corresponding figure legend to clarify the reader.

We also agree with the interpretation of the reviewer that DAP5 may promote re-initiation of translation at uORFs, such as uORF6. We think that re-initiation is not exclusive to the mORF. However, at the moment we are unable to prove this hypothesis, as it requires the development of methodologies/ approaches that monitor the translation of multiple uORFs and main CDS simultaneously. In the future, we would like to develop single-molecule approaches using reporter transcripts containing the 5' leaders of DAP5 targets to have additional insights into the details of DAP5-dependent re-initiation of translation. We have included a sentence in the discussion to highlight the need for additional studies to understand the details of DAP5-dependent translation.

Reviewer 3

We thank the reviewer for reading our manuscript in detail and critically. We understand the points raised by the reviewer and hope that the revised version of the manuscript and our answers clarify most of the raised issues.

Overexpression of a 4E binding protein (4EBP) construct prevents DAP5-dependent translation, suggesting that DAP5 associates with capped mRNAs and is dependent on the eIF4E-eIF4G interaction for its function. This is particularly interesting considering that DAP5 does not have an eIF4E binding domain. It would have been a welcome addition for the authors to further explore how its function relies on the eIF4E-G interaction.

Our data reveals the unexpected observation that DAP5 participates in cap- and eIF4F-dependent translation. Most of the previous work on this non-canonical initiation factor referred that DAP5 is either involved in IRES-dependent translation or eIF3d-dependent translation. This new model of DAP5-dependent initiation then indicates that it is important to understand the relationship and dynamics of distinct initiation complexes during translation of eukaryotic mRNAs. Our data suggests so far that eIF4F-dependent recruitment of PICs is essential for the initiation of translation by DAP5. As DAP5 lacks eIF4E-binding sites, it is unclear how it is recruited to target mRNAs. The data presented in our manuscript describes two features required for DAP5 recruitment: the eIF4F complex and the RNA helicase eIF4A. The latter is a well-established partner of DAP5, whereas the former is not. The structured nature of the 5' leaders of DAP5 targets coupled with the data published by Ingolia and collaborators¹ on cells treated with Rocaglates (drugs that clamp eIF4A to mRNA) indicates that eIF4A binds to DAP5 targets. Our Ribo-Seq in wild type and DAP5-null cells shows that cap-proximal uORF translation in the 5' leaders is DAP5-independent and most likely eIF4F-dependent. In addition, we also observed that eIF4G binds efficiently to DAP5 targets as part of the eIF4F complex (Fig. 2e-g). However, eIF4F (and eIF4G) are insufficient to promote translation of cap-distant ORFs, such as the main CDS. Our interpretation is that the complex nature of the 5' leaders of DAP5 targets limits the availability or the efficiency of the eIF4F-loaded PICs to scan and translate cap-distant ORFs. DAP5 is then the factor that increases the ability of the PICs that scanned and translated uORFs, to continue moving along the long 5' leaders and initiate translation at downstream ORFs. Thus, DAP5 is dependent on the eIF4F

solely because this initiation complex is essential for the initial recruitment of ribosomes to the capped mRNA. These ribosomes are then utilised by DAP5 in the initiation of translation at cap-distant ORFs.

Despite these insights into DAP5 function, the remainder of the paper is plagued by inconclusive experiments which are difficult to interpret in the context of the authors' major claim: that is, that DAP5 is essential for translation re-initiation on uORF-containing mRNAs. The alternative hypothesis which goes unmentioned and untested is that DAP5 could promote leaky scanning. While these experiments might test DAP dependent translation of the mORF, they are not set up to test their hypotheses and are completely confounded by the complexity of the WNK1 5'UTR which contains 8 different uORFs. What would be required to test the alternative hypotheses of reinitiation and leaky scanning would be a simpler 5' UTR with a single uORF at play. A good starting experiment would be to mutate the start codon and show that uORF mediated regulation is lost and that mORF translation becomes DAP5 independent. Mutation of the stop codons in the complicated WNK1 5'UTR may test something related to the importance of uORF length in regulation of mORF translation, but these ideas are not particularly novel (REF). In light of these many limitations, the major claim of the paper that DAP5 promotes translation re-initiation is unsupported by the experiments provided.

We agree with the reviewer that *WNK1* 5' leader is very complex as it contains multiple uORFs and strong secondary structure elements. Our choice to use this DAP5 target as a study case was based on multiple facts:

1. *WNK1* mRNA shows the highest change in TE in DAP5-null cells [\log_2FC TE (DAP5KO vs control) = -2.6], suggesting it was a direct DAP5 target (see Table S1).
2. DAP5 efficiently associated with *WNK1* mRNA (3-fold enrichment in RNA pulldown assays; see Fig. 2e).
3. *WNK1* mRNA and protein levels could be easily determined using qPCR and commercially available antibodies (see Figs. 1d, 2e, S1k).
4. The 5' leader sequences of DAP5 targets are characterized by the presence of multiple uORFs.

We have also followed the reviewer's suggestion and used a reporter mRNA containing a simpler 5' leader sequence. *AKT1* 5' leader contains a single uORF in a reading frame distinct from the main ORF (see Fig. S3b). *AKT1* 5' leader is also sufficient to promote DAP5-

dependent translation of R-LUC (see Fig. 2c). As in the case of the *WNK1* R-LUC reporter (see Fig. 5), we abolished uORF translation by mutating the uORF start codon (*AKT1* R-LUC Δ START). As observed for the *WNK1* reporter (see Fig. 5), uORF translation limited main CDS translation, as R-LUC translation increased 3-fold in the mutated reporter (see figure attached to this text – Fig. II a-c). Importantly and contrary to the expectation, R-LUC translation remained DAP5-dependent (Fig. I a-c). This result suggests that the role of DAP5 in translation is independent of the mutated start codon and translation of the corresponding uORF. That is, DAP5 does not promote skipping of the uORF start site (leaky scanning) by the scanning PICs in order to maximize main ORF translation.

Fig. II. WT and DAP5-null cells were transfected with plasmids expressing *AKT1*-R-LUC reporters, V5-SBP-MBP or V5-SBP-DAP5 and F-LUC-GFP. Following transfection, luciferase activities were measured (a) and mRNA levels were determined by RT-qPCR (b). R-LUC activity and mRNA levels were normalized to the transfection control F-LUC-GFP and set to 100% in WT cells. Protein and mRNA ratios on WT and null cells are depicted in c. Bars show the mean value and error bars indicate the SD (n=3). The immunoblot in d shows the

expression levels of the proteins used in the assay. Membranes were incubated with anti-V5, GFP, and R-LUC antibodies.

Our interpretation is that the inhibitory structured elements present in the 5' leader impose a requirement for DAP5 in the initiation of translation at the main AUG. We envision that scanning in the burdened 5' leader is disturbed and the slow scanning PICs can initiate translation at codons other than the cognate upstream start codon.

To understand if DAP5 promotes leaky scanning or reinitiation of translation after uORF translation, we have also manipulated uORF translation termination by mutating all the termination codons in the *AKT1* 5' leader. In this reporter (*AKT1* R-LUC Δ STOP), leaky scanning can still occur as the uORF start site was not modified. If DAP5 promotes leaky scanning, in WT cells the PIC will scan past the uORF start site and translate R-LUC main ORF. However, we see that R-LUC translation is greatly impaired when termination of translation is not taking place at the 5' leader (Fig. II a-c). Most likely, the complex nature of the 5' leader enforces initiation of translation at the uORF start codon for the majority of the scanning complexes. In a 5' leader devoid of termination codons, the ribosomes translate past the main AUG and R-LUC is not produced. This result further indicates that the PICs are mostly unable to scan past the uORF start codon, even in the presence of DAP5, and that reinitiation of translation following uORF translation must then drive main CDS translation. Together with the observation that uORF translation is pervasive in the 5' leader of DAP5 targets due to the structured nature of the mRNA, we conclude that DAP5 is required for the reinitiation of translation following uORF translation.

As the data regarding the *AKT1* reporter is similar to the results with the *WNK1* reporter, the data is only shown here in the answer to the reviewers.

We are also aware that mutation of the STOP codon will change the length of the uORF and affect the translation efficiency of the main ORF, as shown by Kozak in 2001². We simply intended to show that inhibition of reinitiation by manipulation of uORF length interferes with the ability of DAP5 to promote main CDS translation. This evidence further corroborates the idea that DAP5 function is linked to reinitiation of translation in mRNAs with complex 5' leaders.

Major points:

1. While it is standard in the field to use drugs such as harringtonine and lactidomycin to

identify translation initiation sites, these experiments suffer from a likely overestimation of uORF utilization. What happens in cells treated with these drugs is that all ribosomes run off of all the mRNAs in the cell, thus dramatically increasing the concentration of ribosomes in the cell. In this situation, it seems likely that ribosomes will fill first all the AUG start codons in the cell (and become trapped) and then they will move on to less optimal initiation sequences. This likely explains the very abundant use of non-canonical start codon usage revealed in these experiments. The methods developed by Shu-bin Qian where ribosomes are trapped with drugs at start sites and cleared in lysate by puromycin likely do a better job of capturing the authentic biology of translation initiation.

We agree with the reviewer and have re-analysed the sites of translation initiation in DAP5 targets using quantitative profiling of initiating ribosomes (QTI) data published by Shu-bin Qian³. This analysis confirmed that translation starts at multiple cognate and near-cognate start codons in the 5' leaders of DAP5 targets, as exemplified by *WNK1*. In comparison to all transcripts expressed in the cell, the number of translation initiation events in the 5' leaders of DAP5 targets is increased in cells lacking DAP5 (see Fig. 1e, g of the revised manuscript).

2. It would be useful to use some reporters with uORFs that do not respond to DAP5 as negative controls. While it is nice to see three that emerged as strong candidates from ribosome profiling, it should be easy to find some candidates that share the feature of having a 5' uORF but which behave differently to DAP5.

Our data also shows that multiple RNAs with uORFs in the 5' leaders do not respond to DAP5. One of the evidences indicating that DAP5 is not always required for re-initiation of translation is the fact that DAP5 targets only partial overlap with the targets of DENR, a non-canonical factor shown to support re-initiation after uORF translation (see Fig. 7i and Supplementary Table 3). Among the reinitiation-dependent (DENR-dependent⁴) but DAP5-independent mRNAs are for instance the *ATF4*, *MAP2K6*, *PIK3R1* and *DROSHA* mRNAs. To follow the reviewer's suggestion, we tested if DAP5 was required for *ATF4* translation, which is known to be re-initiation dependent after uORF translation. We exposed wild type and DAP5-null HEK293T cells to thapsigargin-induced endoplasmic reticulum (ER) stress. Thapsigargin-treated cells showed increased eIF2 α phosphorylation. *ATF4* expression was induced in wild type and DAP5-null treated cells (Fig. III, lanes 4 vs 2) and thus DAP5-independent. This data,

presented only in the point-by-point answers due to the limitations in text length and figure number, indicates that DAP5 is not required for re-initiation of ATF4 translation.

Fig. II - ATF4 expression was assessed by immunoblotting in HEK293T cells treated with ethanol (-) or thapsigargin (+). Blots were probed with anti-ATF4, DAP5, eIF2α-P and TUBULIN antibodies. The asterisk (*) indicates a non-specific protein cross-reacting with ATF4 antibody. The ATF4 protein band runs below the non-specific signal and is only observed in the upon induction of ER stress, as expected.

3. Throughout the manuscript, no statistical tests are performed. For example in Figure 4b, p-values would be helpful for the Δ STOP constructs. It is difficult for the reader to reasonably interpret whether the conclusions are correct without statistics to back up the data.

We have followed the suggestion of the reviewer and have added the statistical analysis and p-values to the figures. The analyses support the conclusions stated in the manuscript.

4. In Fig 3D, the authors argue that because a large deletion of the 5'UTR (Δ 2) abolishes RLuc expression, uORFs must be mandatory for translation of the mORF. This experiment just doesn't make sense. First, it goes against a large body of literature suggesting that uORFs are usually inhibitory to translation of the downstream mORF (see PMID: 19372376 for one example). Second, the experiment is set up incorrectly. To determine the importance of uORF translation to mORF translation, the uORF start codons should simply be mutated (i.e. AUG \diamond AAA). Additionally, the fact that the Δ 1 mutation causes mORF expression to go up instead of down argues against the author's general claim that uORF translation is "mandatory" to downstream mORF translation because in this case, deletion of uORFs in this region leads to increased translation from the mORF (lane 3 vs. lane 6).

We agree with the reviewer that our interpretation of the data obtained with the reporter carrying deletions in the 5' leader of *WNKI* mRNA was unclear and misleading. We have rephrased our interpretation of the data. In the revised version of the manuscript we introduce

that the cap-proximal truncations in *WNK1*-R-LUC mRNA position a predicted G-quadruplex (Gq) at the 5' end of the mRNA that most likely prevents 40S subunits from binding to the mRNA, preventing R-LUC translation. Our additional experiments with *WNK1*-R-LUC reporters substituting the structured region of the 5' leader by an unstructured sequence of 18 CAA repeats support the notion that the 5' leaders of DAP5 targets contain regulatory and highly structured RNA elements that restrict translation and impose DAP5-dependence for main CDS translation (see Fig. 3 of the revised manuscript).

5. In Figures 4, 5, and 6 the authors claim that lengthening the respective uORFs causes *WNK1* reporter translation to no longer be dependent on DAP5. However, in all three cases, lengthening the uORF completely abrogates the mORF signal (up to ~100-fold decrease in some cases) – but even at these low levels of expression there appears to remain some DAP5 dependence. Given the low signal for this assay, it could be important to do a control experiment to show that the assay is in the dynamic range.

We have consistently observed that lengthening of uORFs significantly disrupted R-LUC (main ORF) translation. The low levels of R-LUC expression were determined by measuring R-LUC activity, or mRNA and protein levels using northern blotting/ qPCR and Western blotting, respectively. Changes in mRNA levels were not sufficient to explain the decrease in R-LUC expression. In addition, we show that F-LUC activity, mRNA and protein levels are not affected in the same experimental conditions. Thus, we believe that the conditions used to evaluate our results were in the dynamic range of the various assays (we are not sure if this was what the referee is referring to). Moreover, the measurement of luciferase activities were also acquired in cells transfected with the amount of reporter DNA that produces signals above background level and that do not exceed the linear range of the detection method.

Regarding the fact that even at low levels the reporters still seem to show some DAP5 dependence, the error bars associated with the different measures overlap in the majority of the conditions, and thus may indicate that the remaining R-LUC translation does not significantly vary between cells with and without DAP5.

6. In Figures 4C and 5C the FLuc expression is consistently decreased in every single DAP5-null line. Presumably, the FLuc construct should be a transfection control and should not

encode a DAP5-dependent 5'UTR or uORFs. The observation that FLuc levels are consistently responding to DAP5 knockout is troublesome.

Indeed, the F-LUC reporter was used as a transfection control in our experiments. The 5' leader of the F-LUC reporter is short and lacks uORFs. In all the experiments it was important to add this reporter control because we have systematically observed that transfection efficiency is lower in cells lacking DAP5. DAP5-null cells proliferate less than the corresponding WT cells, most likely due to changes in the TE of mRNAs encoding kinases and phosphatases that participate in signalling cascades that support cell proliferation. Thus, even if we used more DAP5-null cells per plate (see Material and Methods section Transfection, northern and western blotting) and transfected the same amount of the control reporter, the expression of F-LUC (or mCherry reporter, see for instance Supplementary Fig. 7d) was generally decreased in the null cells. Normalizing the R-LUC activity, mRNA levels and protein levels by the respective F-LUC levels, thus allows to take in account changes in transfection efficiency in the different experiments.

7. The conclusions from Figure 5D may be misinterpreted. The fact that DAP5-KO causes N-terminally-extended RLuc expression to increase and mORF expression to decrease (compare lanes 3 and 4) could support a leaky scanning model (i.e. if loss of DAP5 prevents scanning past these extended uORFs).

The experiment from Fig. 5D of the original manuscript shows that N-terminally extended R-LUCs are synthesised when termination of uORF translation is altered by removing STOP codons in the 5' leader. We are in the opinion that the start codons utilized by the scanning complexes to translate these extended versions of R-LUC are recognized also in the WT reporter, where the STOP codons remain. The main difference is that in the WT reporter, uORF translation originates a small peptide/protein that is invisible in the experiment because it is not tagged and terminates before the main ORF. In the reporter lacking the STOP codons, initiation of translation at an upstream start codon that is in the same reading frame as the main AUG, originates an R-LUC protein that we are able to detect using anti-R-LUC antibodies and luciferase activity measurements. We do understand the point raised by the reviewer that the presence of extended R-LUC proteins might suggest that the PICs are unable to scan past the uORF start codons in the absence of DAP5. However, this experiment and others in the

manuscript do not support this conclusion. Here are some of the reasons why we believe that DAP5 does not assist the PICs to scan past the uORF initiation codons to guarantee main CDS translation, but is actually promoting re-initiation following uORF translation:

a) In Fig. 5d of the original manuscript, the N-terminally extended versions of R-LUC are present in WT and null cells, i.e., uORF translation occurs in the presence and absence of DAP5.

b) In Fig. 6 of the revised manuscript in which we simultaneously monitor uORF and CDS translation using a split-fluorescent approach, we observe that uORF and main CDS are simultaneously translated and only main CDS expression was DAP5-dependent.

c) uORF translation is pervasive in DAP5 targets as seen in the ribosome profiling data due to the intricate nature of the RNA.

d) In the revised manuscript, we have also tested main CDS translation of a reporter mRNA in which all cognate and near-cognate initiation codons upstream of the main AUG were altered (Δ START). In the absence of recognizable uORF start codons, translation of R-LUC remained DAP5-dependent. We envision that the highly structured nature of the transcript strongly compromises main CDS translation. This strong inhibitory element imposes strict DAP5-dependency. This result suggests that DAP5 may play an important role in melting strong secondary structures in the 5' leader. As the structured nature of the 5' leader interferes with PIC scanning, we envisioned the possibility that initiation of translation can still occur at codons other than the cognate and near-cognate start codons mutated in the 5' leader, as observed in about 22% of DAP5 targets (Fig. 1j). Thus, to further test that re-initiation of translation is pervasive in the 5' leader of DAP5 targets, we also modified all STOP codons in the 5' leader. As explained above (see page 14), we observed that termination of translation in the 5' leader was required for main CDS translation, further supporting the model proposed in our manuscript that DAP5 promotes re-initiation of translation.

8. The experiments in Figure 7 show modest effects and it is not clear how they support the overall conclusions of the manuscript. Perturbation of termination/recycling could certainly have an impact on re-initiation but the data are just not very compelling.

To further corroborate that termination after uORF translation leads to main CDS translation, we also interfere with termination and recycling by manipulation of the expression of key translation factors. Because changes in levels of these factors (e.g. ABCE1) can affect cell viability, we have performed the assays in conditions that keep the cells alive and dividing (e.g.

partial knockdowns - Fig. 7d). In these conditions, we observed significant effects on the expression of re-initiation dependent reporters (now supported by statistical analysis). Altogether, the results further corroborate that in DAP5 targets termination of translation after uORF translation is required to the initiation of translation at the main CDS.

References

- 1 Iwasaki, S., Floor, S. N. & Ingolia, N. T. Rocaglates convert DEAD-box protein eIF4A into a sequence-selective translational repressor. *Nature* **534**, 558-561, doi:10.1038/nature17978 (2016).
- 2 Kozak, M. Constraints on reinitiation of translation in mammals. *Nucleic Acids Res* **29**, 5226-5232, doi:10.1093/nar/29.24.5226 (2001).
- 3 Gao, X. *et al.* Quantitative profiling of initiating ribosomes in vivo. *Nat Methods* **12**, 147-153, doi:10.1038/nmeth.3208 (2015).
- 4 Bohlen, J. *et al.* DENR promotes translation reinitiation via ribosome recycling to drive expression of oncogenes including ATF4. *Nat Commun* **11**, 4676, doi:10.1038/s41467-020-18452-2 (2020).

REVIEWER COMMENTS

Reviewer #1 (Remarks to the Author):

The authors have done a nice job of addressing the concerns raised in the original review, and I strongly support publication of this interesting study.

Reviewer #2 (Remarks to the Author):

In this manuscript, authors conclude that DAP5 promotes translation re-initiation in mRNAs with complex 5' leader sequences.

Although the revised manuscript is significantly improved, I still have one issue, which would be good to address prior to publication. In Fig 5d, authors claim "Our results are consistent with a re-initiation mechanism in which DAP5 promotes initiation by reusing ribosomal complexes previously engaged in translation." However, in Fig 5d, we observe that expressing V5-SBP-DAP5 significantly increases translation efficiency of Δ START/STOP-R-Luc (although not enough to be observed by Western blot; Fig 5e), construct that does not present any uORF. This point should be better addressed before the authors conclude that DAP5 promotes re-initiation by reusing ribosomal complexes previously engaged in translation.

Reviewer #3 (Remarks to the Author):

In this manuscript, Weber et al. interrogate the role of DAP5 (eIF4G2) in promoting main open reading frame (mORF) translation of certain mRNAs.

As in the first round of review, we remain convinced that DAP5 promotes mORF translation of certain mRNAs with structured, uORF-containing leader sequences. The DAP5 deletion experiments remain a useful characterization demonstrating that all domains of the protein are essential for promoting mORF translation of its target mRNAs. Additionally, the authors have now demonstrated that DAP5 does not function through leaky scanning. In general, the data are convincing for these general statements and we feel that the manuscript would be of greater value if it were to limit its focus to these conclusions.

Despite these merits, much of the manuscript still contains experiments that have multiple possible interpretations. The authors appear to be adamant in arguing for a re-initiation model despite the fact that no experiment directly tests this model. The fact that long uORFs are more inhibitory to translation could be rationalized by the idea that re-initiation is less efficient on long uORFs or by an equally compelling argument that 48S scanning rates are considerably faster than 80S elongation rates (see BioRxiv preprint by Wang and Puglisi); therefore, the loading and scanning of a new 48S is going to be more impaired by an 80S ribosome slowly elongating through a long uORF than through a short uORF. Similarly, the fact that impairing termination more severely decreases translation from uORF-containing WNK1 constructs than non-uORF containing constructs (Figure 7) could argue for a re-initiation model or could simply be explained by the idea that ribosomes trapped on uORF stop codons due to a recycling defect act as a roadblock for new scanning 48S pre-initiation complexes.

Furthermore, some data presented in the manuscript actually argue against the re-initiation model being correct. For example, the Δ START experiments referenced above, in addition to ruling out leaky scanning, are also inconsistent with re-initiation since mRNAs which lack uORFs should no longer be DAP5-dependent if re-initiation of uORFs is the salient function being provided by DAP5. Instead, as Reviewer 1 suggests, 5'UTR structure appears to be the most important feature for DAP5-dependence as the Δ struct mutants in Figure 3i no longer depend on DAP5 for their translation despite still

containing uORFs.

Taken together, we do feel that this manuscript offers insights into DAP5 function that are of value and should be published. Nonetheless we feel that a simpler version of the manuscript which focuses on the general conclusions directly supported by the data would be stronger and altogether less confounding to the field.

NCOMMS-21-06391A

We would very much like to thank the referees once more for taking the time to review our work. We appreciate their comments and have addressed the points raised by each reviewer. Please find below an outline of the changes made to the manuscript (in blue) to address the different comments (in black).

Reviewer 1

The authors have done a nice job of addressing the concerns raised in the original review, and I strongly support publication of this interesting study.

We thank the reviewer for their support.

Reviewer 2

In this manuscript, authors conclude that DAP5 promotes translation re-initiation in mRNAs with complex 5' leader sequences. Although the revised manuscript is significantly improved, I still have one issue, which would be good to address prior to publication. In Fig 5d, authors claim “Our results are consistent with a re-initiation mechanism in which DAP5 promotes initiation by reusing ribosomal complexes previously engaged in translation”. However, in Fig 5d, we observe that expressing V5-SBP-DAP5 significantly increases translation efficiency of Δ START/STOP-R-Luc (although not enough to be observed by Western blot; Fig 5e), construct that does not present any uORF. This point should be better addressed before the authors conclude that DAP5 promotes re-initiation by reusing ribosomal complexes previously engaged in translation.

We thank the reviewer for their positive comments.

The reviewer's observation is correct. The *WNK1* Δ START/STOP reporter, and other reporters where translation of the main CDS is independent of DAP5 (see also Fig. 3i), show increased TE following the overexpression of V5-SBP-DAP5 in the DAP5-null cells. These results suggest that DAP5 may also promote translation in transcripts in the absence of uORF translation or structured RNA elements. To account for these observations, we have modified the text (please see page 13). We also note that we are unable to state that this alternative function of DAP5 is present in cells as the modified transcripts are highly distinct from the natural DAP5 targets.

Reviewer 3

As in the first round of review, we remain convinced that DAP5 promotes mORF translation of certain mRNAs with structured, uORF-containing leader sequences. The DAP5 deletion experiments remain a useful characterization demonstrating that all domains of the protein are essential for promoting mORF translation of its target mRNAs. Additionally, the authors have now demonstrated that DAP5 does not function through leaky scanning. In general, the data are convincing for these general statements and we feel that the manuscript would be of greater value if it were to limit its focus to these conclusions.

We thank the reviewer for carefully reading our revised manuscript. We appreciate the issues raised by the reviewer and in this version of the manuscript we have limited some of the conclusions and focused mainly on the data. We believe that we cannot exclude the hypothesis that DAP5 stimulates main CDS translation in re-initiation-dependent transcripts. However, we have now also discussed other interpretations of the data and highlighted the role of DAP5 and eIF4A on mRNAs with structured 5' leaders.

Despite these merits, much of the manuscript still contains experiments that have multiple possible interpretations. The authors appear to be adamant in arguing for a re-initiation model despite the fact that no experiment directly tests this model. The fact that long uORFs are more inhibitory to translation could be rationalized by the idea that re-initiation is less efficient on long uORFs or by an equally compelling argument that 48S scanning rates are considerably faster than 80S elongation rates (see BioRxiv preprint by Wang and Puglisi); therefore, the loading and scanning of a new 48S is going to be more impaired by an 80S ribosome slowly elongating through a long uORF than through a short uORF. Similarly, the fact that impairing termination more severely decreases translation from uORF-containing WNK1 constructs than non-uORF containing constructs (Figure 7) could argue for a re-initiation model or could simply be explained by the idea that ribosomes trapped on uORF stop codons due to a recycling defect act as a roadblock for new scanning 48S pre-initiation complexes.

We agree with the reviewer that the elongating or trapped ribosomes can limit and constitute roadblocks to the 43S complexes that scanned past the start codon of the uORF. However, we do not favour this interpretation of the data because of multiple observations.

- 1) As shown in Fig. 5, the leaky scanning model does not apply to the reporters used in our study.

2) To specifically prevent leaky scanning and guarantee initiation at the uORF start site, in the experiments addressing the effect of uORF length on DAP5-dependent translation, we have used reporters where uORF translation was driven by a natural AUG (uORF6, Fig. 4) or an altered AUG start codon (uORF2+, Figs. 7 and S6). In the latter, conversion of GUG to AUG (Fig. S6), did not alter the translatability of the mRNA as it mimicked the reporter containing the natural 5' leader. These reporters allowed us to study DAP5 function in conditions where leaky scanning is suppressed.

3) Puglisi and co-workers (BioRxiv preprint) have elegantly measured the scanning rate of 43S complexes under normal conditions in unstructured 5' leaders. They have also demonstrated that 5' UTR secondary structures modulate scanning dynamics. Although still unknown, we envision that the scanning rate of the 43S complexes in the 5' leaders of DAP5 targets is highly disturbed by the strong structured RNA elements.

Based on these reasons, it seems unlikely that scanning complexes will meet and be blocked by elongating or trapped ribosomes, as recognition of cognate and near-cognates start codons will lead to the formation of an 80S ribosome and initiation of translation. Therefore, we tend to favour the hypothesis that DAP5 enhances main CDS translation following uORF translation, and thus acts on transcripts where re-initiation of translation must occur to drive main CDS translation. Nevertheless, because we are currently unable to simultaneously measure uORF(s) and main CDS translation in a single mRNA, and the dynamics of 43S and 80S complexes on the transcript, we are unable to completely rule out the possibility that the limited/disturbed 43S scanning rates due to the slower 80S elongating or trapped ribosomes inhibit the function of DAP5 in the *WNK1* reporters. Therefore, we have added such interpretation to the manuscript.

Furthermore, some data presented in the manuscript actually argue against the re-initiation model being correct. For example, the Δ START experiments referenced above, in addition to ruling out leaky scanning, are also inconsistent with re-initiation since mRNAs which lack uORFs should no longer be DAP5-dependent if re-initiation of uORFs is the salient function being provided by DAP5. Instead, as Reviewer 1 suggests, 5'UTR structure appears to be the most important feature for DAP5-dependence as the Δ struct mutants in Figure 3i no longer depend on DAP5 for their translation despite still containing uORFs.

We agree with the reviewers that DAP5 and eIF4A function is intrinsically associated with the structured nature of the target transcripts, and have further highlighted this idea in the

manuscript. In line with the observations of Wang and Puglisi (BioRxiv preprint) regarding the modulation of scanning rates by RNA structures in 5' leaders, DAP5 most likely recruits and/or stimulates eIF4A activity in order to overcome the energy barriers faced by the scanning complex in the highly structured 5' leaders. In addition, since DAP5 is known to bind to multiple eIFs (e.g. eIF3, eIF2) it may also promote 40S loading and binding to the mRNA.

The Δ structure reporter used in Fig. 3 clearly indicates that a transcript lacking structured RNA elements but still containing uORFs does not require DAP5 for initiation of translation at the main CDS. Although this reporter confirms the intricate relationship between DAP5 and RNA structure, we can't use it to study the influence of uORF translation on DAP5 function. As shown for instance by Puglisi and collaborators (BioRxiv preprint), the efficiency of near-cognate translation at the uORFs in an unstructured 5' leader is low. Consequently, the majority of the 43S complexes will bypass the uORFs and scan until the main AUG without requiring DAP5 and eIF4A. Thus, the Δ structure reporter is most likely also devoid of uORF translation as the 43S scanning rate is not limited.

With the Δ START and Δ STOP WNK1 reporters we explored the requirement of uORF translation on DAP5-dependent initiation. In the absence of the experimentally determined START codons in the 5' leader, main CDS translation was still DAP5-dependent. The simplest explanation is that DAP5 and eIF4A resolved the energetic barriers along the structured 5' leader until the 43S complexes reach the main AUG. In the absence of translation termination in the 5' leader, we only observed DAP5-independent translation of long protein products, such as the N-terminally extended versions of R-LUC. This result indicated that even in the presence of DAP5, the 43S complexes mainly initiated translation at the upstream start codons and were unable to reach the main AUG in the absence of termination in the 5' leader. When the 5' leader was devoid of START and STOP codons, translation at the main AUG remained inhibited. Thus, the hypothesis that DAP5 together with eIF4A resolves the constraints imposed by RNA structure and leads the scanning complexes towards the main AUG is no longer valid. We then had to assume that the limited scanning rate of the 43S complexes in the structured 5' leader enhanced the initiation of translation at codons other than the experimentally determined AUG and near-cognates codons, i.e., uORF translation still occurred at 5' leader of the Δ START reporter. In that case, only the inhibition of termination in the 5' leader prevented initiation at the main AUG, suggesting that main CDS translation is preceded by an upstream termination

event. Altogether, the experiments are in favour of a model in which DAP5 is required for main CDS translation following uORF translation in structured 5' leaders. In other words, it is the combination of structured RNA elements and uORF translation that define a DAP5-dependent mRNA.

We appreciate the discussion, arguments and the different data interpretations provided by the reviewer. These have helped us to improve and clarify issues on the manuscript. Although we are open to include other data interpretations and tone down some conclusions in the revised manuscript, we believe that we cannot ignore the compelling arguments that support the re-initiation model.